# Long-range Meta-path Search through Progressive Sampling on Large-scale Heterogeneous Information Networks

## Abstract

Utilizing long-range dependency, though extensively studied in homogeneous graphs, is rarely studied in large-scale heterogeneous information networks (HINs), whose main challenge is the high costs and the difficulty in utilizing effective information. To this end, we investigate the importance of different meta-paths and propose an automatic framework for utilizing long-range dependency in HINs, called *Long-range Meta-path Search through Progressive Sampling* (LMSPS). Specifically, to discover meta-paths for various datasets or tasks without prior, we develop a search space with all target-node-related meta-paths. With a progressive sampling algorithm, we dynamically shrink the search space with hop-independent time complexity, leading to a compact search space driven by the current HIN and task. Utilizing a sampling evaluation strategy as the guidance, we conduct a specialized and expressive meta-path selection. Then, we can employ a part of effective meta-paths instead of the full meta-path set to reduce costs. Extensive experiments on eight heterogeneous datasets demonstrate that LMSPS discovers effective long-range meta-paths and outperforms state-of-the-art models. Besides, it ranks top-1 on the leaderboards of `ogbn-mag` in Open Graph Benchmark.

## 1 Introduction

Heterogeneous information networks (HINs) are widely used for abstracting and modeling multiple types of entities and relationships in complex systems by various types of nodes and edges. For example, the large-scale academic network, `ogbn-mag`, contains multiple node types, *i.e.*, Paper (P), Author (A), Institution (I), and Field (F), as well as multiple edge types, such as Author $\xrightarrow{\text{writes}}$ Paper, Paper$\xrightarrow{\text{cites}}$Paper, Author $\xrightarrow{\text{is affiliated with}}$ Institution, Paper$\xrightarrow{\text{has a topic of}}$Field. These elements can be combined to build higher-level semantic relations called meta-paths (Sun et al., 2011). For instance, PAP is a 2-hop meta-path, and PFPAPFP is a 6-hop meta-path related to long-range dependency.

Utilizing long-range dependency is an essential issue in the graph field. For homogeneous graphs, many deep graph neural networks (Li et al., 2019; Bianchi et al., 2020; Alon & Yahav, 2021; Wei et al., 2023) have been developed to gain benefit from long-range dependency. Utilizing long-range dependency is also crucial for HINs. For example, the Internet movie database (`IMDB`) contains $21K$ nodes with only $87K$ edges. Such sparsity means each node has only a few directly connected neighbors and requires models to enhance the node embedding from long-range neighbors.

Traditional HGNNs can be classified into two categories: metapath-free methods and metapath-based methods. Metapath-free HGNNs (Zhu et al., 2019; Hong et al., 2020; Hu et al., 2020; Lv et al., 2021) learn the node representations by utilizing information from $l$-hop neighborhoods through stacking $l$ layers. However, on large-scale HINs, the number of nodes in the receptive field grows exponentially with the number of layers. So leveraging large-hop neighborhoods incurs high computation and memory costs during training, making this category of HGNNs hard to expand to large hops.

Metapath-based HGNNs (Ji et al., 2021; Yang et al., 2023; Wang et al., 2019; Fu et al., 2020) obtain information from $l$-hop neighborhoods by utilizing single-layer structures and meta-paths with the maximum hop $l$, *i.e.*, all meta-paths are no more than $l$ hops. However, to increase the receptive field, the maximum hop needs to be large enough because there are no stacking layers. Some

methods (Wang et al., 2019; Ji et al., 2021; Fu et al., 2020) employ manually-designed meta-paths, which are typically short and highly challenging to be obtained in various schema-rich or large-scale HINs. Other methods (Yun et al., 2019; Li et al., 2021b; Yang et al., 2023) utilize all target-node-related meta-paths instead of manual ones. However, the number of target-node-related meta-paths also grows exponentially with the maximum hop value, so using a large maximum hop also requires high costs in these methods. For instance, in SeHGNN (Yang et al., 2023), the maximum hop in `ogbn-mag` (Hu et al., 2021) is set to a small value of 2, which is insufficient for large-scale datasets. Therefore, it has not been well addressed yet on how HGNNs can leverage long-range dependencies efficiently, especially on large-scale datasets.

In this paper, we investigate the importance of different meta-paths and gain two observations: (1) A few meta-paths dominate the performance, and (2) some meta-paths negatively impact the performance. The second observation explains why few HGNNs can benefit from long-range neighbors, *i.e.*, messages in HINs could be noisy or redundant to each other, and long-range dependencies increase the difficulty of excluding negative information. Moreover, the two findings earn the opportunity to leverage long-range dependencies by keeping a small part of effective meta-paths.

Motivated by the above observations, we develop a novel method, termed Long-range Meta-path Search through Progressive Sampling (LMSPS), to effectively utilize the information from neighbors at different distances in HINs. Specifically, LMSPS first builds a comprehensive search space with all target-node-related meta-paths, and then shrinks the search space with a progressive sampling strategy. Finally, LMSPS selects the top-$k$ meta-paths on the compact search space based on the results of sampling evaluation. Consequently, the search stage reduces the exponentially increased number of meta-paths to a constant for retraining. Experiments on four widely-used real-world datasets and four manual sparse large-scale datasets demonstrate that LMSPS outperforms state-of-the-art models for node classification on HINs, especially on sparse HINs.

Our main contributions are summarized as follows: (1) We propose a novel meta-path search framework, called Long-range Meta-path Search through Progressive Sampling (LMSPS), which is the first to utilize long-range dependency in large-scale HINs. (2) To search for effective meta-paths, we introduce a progressive sampling algorithm to shrink the search space dynamically and a sampling evaluation strategy for meta-path selection. (3) To our knowledge, LMSPS is the first HGNN whose discovered meta-paths are effective enough to enable other HGNNs to achieve better performance. (4) Thorough experiments demonstrate that LMSPS achieves state-of-the-art performance.

## 2 PRELIMINARIES

**Heterogeneous information network (HIN)** (Sun & Han, 2012). *An HIN is defined as $\mathcal{G} = \{\mathcal{V}, \mathcal{E}, \mathcal{T}, \mathcal{R}, f_{\mathcal{T}}, f_{\mathcal{R}}\}$ with $|\mathcal{T}|+|\mathcal{R}|>2$, where $\mathcal{V}$ denotes the set of nodes, $\mathcal{E}$ denotes the set of edges, $\mathcal{T}$ is the node-type set and $\mathcal{R}$ the edge-type set. Each node $v_i \in \mathcal{V}$ is maped to a node type $f_{\mathcal{T}}(v_i) \in \mathcal{T}$ by mapping function $f_{\mathcal{T}} : \mathcal{V} \to \mathcal{T}$. Similarly, each edge $e_{t \leftarrow s} \in \mathcal{E}$ ($e_{ts}$ for short) is mapped to an edge type $f_{\mathcal{R}}(e_{ts}) \in \mathcal{R}$ by mapping function $f_{\mathcal{R}} : \mathcal{E} \to \mathcal{R}$.*

**Meta-path** (Sun et al., 2011). *A meta-path $P$ is a composite relation that consists of multiple edge types,* i.e., *$P \triangleq c_1 \xleftarrow{r_{12}} c_2 \cdots c_{l-1} \xleftarrow{r_{(l-1)l}} c_l$ ($P = c_1 c_2 \cdots c_l$ for short), where $c_1, \ldots, c_l \in \mathcal{T}$ and $r_{12}, \ldots, r_{(l-1)l} \in \mathcal{R}$.*

A meta-path $P$ corresponds to multiple meta-path instances in the underlying HIN. In HGNNs, using meta-paths means selectively aggregating neighborhood information. The number of meta-paths grows exponentially with the increase of the maximum hop. For example, on `ogbn-mag`, if we want to get a 3-hop meta-path based on the 2-hop meta-path PAP, the next node type has three choices, *i.e.*, A, P, and F. Consequently, for each additional hop, the number of possible meta-paths increases exponentially.

## 3 RELATED WORKS

**Heterogeneous Graph Neural Networks.** Heterogeneous graph neural networks (HGNNs) are proposed to learn rich and diverse semantic information on HINs with different types of nodes or edges. Several HGNNs (Li et al., 2021a; Ji et al., 2021; Lv et al., 2021) have involved high-order

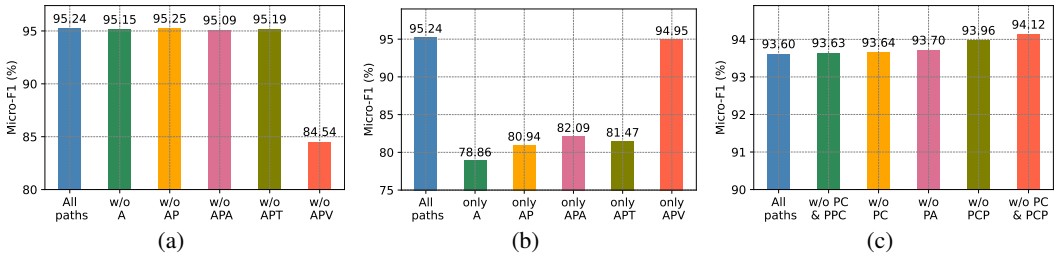

Figure 1: Analysis of the importance of different meta-paths. (a) illustrates the results after removing a single meta-path on `DBLP`. (b) shows the performance of utilizing a single meta-path on `DBLP` (c) illustrates the performance after removing a part of meta-paths on `ACM`.

semantic aggregation. However, their methods are not applied to large-scale datasets due to relatively high costs. Additionally, many HGNNs (Yun et al., 2019; Hu et al., 2020; Wang et al., 2019; Ji et al., 2021) have implicitly learned meta-paths by attention. However, few work employs the discovered meta-paths for the final results, let alone generalize them to other HGNNs to demonstrate the effectiveness. For example, GTN (Yun et al., 2019) and HGT (Hu et al., 2020) only list the discovered meta-paths. HAN (Wang et al., 2019), HPN (Ji et al., 2021) and MEGNN (Chang et al., 2022) validate the importance of discovered meta-paths by experiments not directly associated with the learning task. GraphMSE (Li et al., 2021b) is the only work that shows the performance of the discovered meta-paths. However, they are not as effective as the full meta-path set. Consequently, their learned meta-paths are not effective enough. Unlike these methods, the meta-paths searched by LMSPS not only enable it to outperform existing HGNNs but also are effective on other HGNNs.

**Meta-structure Search on HINs.** Recently, some works have attempted to utilize neural architecture search (NAS) to discover meta-structures. GEMS (Han et al., 2020) is the first NAS method on HINs, which utilizes an evolutionary algorithm to search meta-graphs for recommendation tasks. DiffMG (Ding et al., 2021) searches for meta-graphs by a differentiable algorithm to conduct an efficient search. PMMM (Li et al., 2023) performs a stable search to find meaningful meta-multigraphs. However, meta-path-based HGNNs are mainstream methods (Schlichtkrull et al., 2018; Zhang et al., 2019; Wang et al., 2019; Fu et al., 2020), while meta-graph-based HGNNs are niche. So, their searched meta-graphs are extremely difficult to generalize to other HGNNs. RL-HGNN (Zhong et al., 2020) proposes a reinforcement learning (RL)-based method to find meta-paths. On recommendation tasks, RMS-HRec (Ning et al., 2022) also proposes an RL-based meta-path selection strategy to discover meta-paths. However, how to search meta-structures on large-scale HINs is a non-trivial problem, and how to search meta-structures that are effective across different HGNNs is also challenging. All the above works have not addressed both problems.

## 4 MOTIVATION OF META-PATH SEARCH

Many HGNNs (Yun et al., 2019; Li et al., 2021b; Chang et al., 2022; Yang et al., 2023) employ all the target-node-related semantic information without analyzing its effects. In this section, we conduct an in-depth and detailed study of different meta-paths and gain two crucial conclusions through experiments, helping us to propose the key idea to leverage long-range dependencies effectively.

SeHGNN (Yang et al., 2023) employs attention mechanisms to fuse all the target-node-related meta-paths and outperforms the state-of-the-art HGNNs. We analyze the importance of different meta-paths for SeHGNN on two widely-used real-world datasets `DBLP` and `ACM` from HGB (Lv et al., 2021). All results are the average of 10 times running with different random initializations. For exploratory experiments, we set the maximum hop $l = 2$ for ease of illustration. Then the meta-path sets are {A, AP, APA, APT, APV} on `DBLP`, and {P, PA, PC, PP, PAP, PCP, PPA, PPC, PPP} on `ACM`.

In each experiment on `DBLP`, we remove one meta-path and compare the performance with the result of leveraging the full meta-path set to analyze the importance of the removed meta-path. As shown in Figure 1 (a), removing either A or AP or APA or APT has little impact on the final performance. However, removing APV results in severe degradation in performance, demonstrating APV is the critical meta-path on `DBLP` when $l = 2$. We further retain one meta-path and remove all other

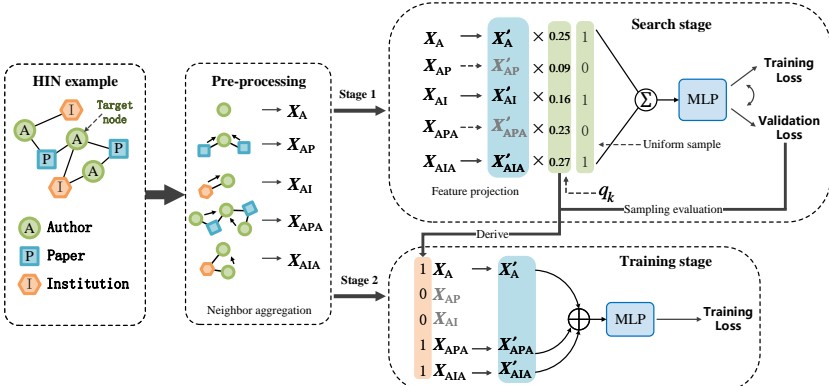

Figure 2: The overall framework of LMSPS. In the search stage, architecture parameters calculating the path strength $q_k$ are alternatively trained with network weights. Based on the progressive sampling algorithm and sampling evaluation strategy in the search stage, the training stage employs $M$ effective meta-paths instead of the full $K$ target-node-related meta-path set.

meta-paths in Figure 1 (b). The performance of utilizing APV is only slightly degraded compared to the full meta-paths set. Consequently, we obtain a finding that a few meta-paths provide major contributions.

In each experiment on ACM, we remove a part of the meta-paths to analyze their impact on the final performance. Results in Figure 1 (c) show that the performance of SeHGNN improves after removing a part of meta-paths. For example, after removing PC and PCP, the Micro-F1 scores improve by $0.52\%$. So, we can conclude that some meta-paths have negative impacts on the performance, which explains why most HGNNs use a maximum hop of 2, *i.e.*, it is difficult to exclude negative information under a larger maximum hop.

Motivated by the above findings, we can employ a part of effective meta-paths instead of the full meta-path set without sacrificing the model performance when the maximum hop is large. Although the number of target-node-related meta-paths grows exponentially with the maximum hop, the proportion of effective meta-paths is small, avoiding computation on numerous noisy or redundant meta-paths.

## 5 THE PROPOSED METHOD

The key point of LMSPS is to utilize a search stage to reduce the exponentially increased meta-paths to a few effective meta-paths. As the search stage aims to determine the relative importance of meta-paths rather than improve the accuracy, its time complexity under large maximum hops can be reduced by progressive sampling. Figure 2 illustrates the overall framework of LMSPS which consists of a super-net in the search stage and a target-net in the training stage. We first introduce the super-net and target-net, and then compare the time complexity of LMSPS with existing HGNNs.

### 5.1 META-PATH SEARCH

The super-net aims to discover effective meta-paths across different HGNNs, so the search results should not be affected by specific modules. Based on this consideration, we develop an MLP-based architecture for meta-path search, consisting of five blocks: neighbor aggregation, feature projection, progressive sampling search, sampling evaluation, and MLP.

Let $\mathbb{P} = \{P_1, \cdots, P_k, \cdots, P_K\}$ be search space with $K$ target-node-related meta-paths, $\boldsymbol{X}^{c_i}$ be the raw feature matrix of all nodes belonging to type $c_i$, and $\hat{\boldsymbol{A}}_{c_i, c_{i+1}}$ be the row-normalized adjacency matrix between node type $c_i$ and $c_{i+1}$. The neighbor aggregation block follows SeHGNN (Yang et al., 2023), which is executed only once in the pre-processing step. It uses the multiplication of adjacency matrices to efficiently calculate the final contribution weight of each metapath-based neighbor to targets. The neighbor aggregation process of $l$-hop meta-path $P_k = cc_1c_2 \ldots c_l$ is:

$$\boldsymbol{X}_k = \hat{\boldsymbol{A}}_{c,c_1} \hat{\boldsymbol{A}}_{c_1,c_2} \cdots \hat{\boldsymbol{A}}_{c_{l-1},c_l} \boldsymbol{X}^{c_l}, \tag{1}$$

where $\boldsymbol{X}_k$ is the feature matrices of meta-path $P_k$. When $l = 0$, $X_k = X^{c_l}$, *i.e.*, the raw feature of the target nodes. Then, an MLP-based feature projection block is used to project different feature matrices into the same dimension, namely, $\boldsymbol{X}'_k = \mathrm{MLP}_k(\boldsymbol{X}_k)$.

To automatically discover meaningful meta-paths for various datasets or tasks without prior, our search space contains all target-node-related meta-paths, severely challenging the efficiency and effectiveness of super-net training. To overcome the efficiency challenge, LMSPS utilizes a progressive sampling algorithm by sampling a part of meta-paths in each iteration and progressively shrinks the initial sampling space (search space). The progressive sampling strategy can overcome the deep coupling issue (Guo et al., 2020) and contribute to a more compact search space specifically driven by the current HIN and task, leading to a more effective meta-path discovery.

Let $\boldsymbol{\alpha} = \{\alpha_1, \cdots, \alpha_k, \cdots, \alpha_K\} \in \mathbb{R}^K$ be corresponding architecture parameters of candidate meta-paths $\mathbb{P}$. We use a Gumbel-softmax (Maddison et al., 2016; Dong & Yang, 2019) over architecture parameters $\alpha_k$ to weight the semantic information of different meta-paths:

$$q_k = \frac{\exp\left[\left(\alpha_k + u_k\right)/\tau\right]}{\sum_{j=1}^{K} \exp\left[\left(\alpha_j + u_j\right)/\tau\right]}, \tag{2}$$

where $q_k$ is the path strength, which represents the relative importance of meta path $P_k$. $u_k = -\log\left(-\log\left(U\right)\right)$ where $U \sim \mathrm{Uniform}(0, 1)$, and $\tau$ is the temperature controlling the continuous relaxation's extent. When $\tau$ is closer to 0, the weights would be closer to discrete one-hot values.

Let $\widetilde{q}_n$ be the $n$-th largest path strength of $\mathbb{Q} = \{q_1, \cdots, q_k, \cdots, q_K\}$, it can be defined as:

$$\widetilde{q}_n = \begin{cases} \max \mathbb{Q} & n = 1 \\ \max\{q | q \in \mathbb{Q}, q < \widetilde{q}_{n-1}\} & n > 1 \end{cases}. \tag{3}$$

We set the number of searched meta-paths to be $M$. Generally, $K \gg M$ under a large maximum hop. For a small maximum hop, if $K \leq 2M$, we do not shrink the search space because it is small enough. If $K > 2M$, less important meta-paths will be dropped progressively with the magnitudes of the path strength as the metric, and the search space size will be shrunk from $K$ to $2M$ during training. The dynamic search space maintained during training can be formulated as:

$$\mathbb{S}_C = \{k | q_k \geq \widetilde{q}_C, \forall 1 \leq k \leq K\} \quad \text{where } C = \begin{cases} \lceil \lambda(K - 2M) \rceil + 2M & K > 2M \\ K & otherwise \end{cases}. \tag{4}$$

Here $C$ is the search space size, $\mathbb{S}_C$ consists of the indexes of retained meta-paths. $\lambda \in [0, 1]$ is a small parameter controlling the number of retrained meta-paths and decreases from 1 to 0 as the epoch increases, and $\lceil \cdot \rceil$ indicates the rounding symbol. When $\lambda = 1$, $C = K$. When $\lambda = 0$, $C = 2M$

As the search stage aims to determine top-$M$ meta-paths, we sample $M$ meta-paths from dynamic search space in each iteration. Therefore, the search cost is relevant to $M$ instead of $K$. The forward propagation can be expressed as:

$$\boldsymbol{Y} = \mathrm{MLP}\Big(\sum_{k \in \mathbb{S}} q_k \cdot \mathrm{MLP}_k(\boldsymbol{X}_k)\Big) \quad \text{where } \mathbb{S} = \mathrm{UniformSample}(\mathbb{S}_C, M). \tag{5}$$

Here $\mathrm{UniformSample}(\mathbb{S}_C, M)$ indicates a set of $M$ elements chosen randomly from set $\mathbb{S}_C$ without replacement via a uniform distribution.

The parameter update in the super-net involves a bilevel optimization problem (Anandalingam & Friesz, 1992; Colson et al., 2007; Xue et al., 2021).

$$\min_{\boldsymbol{\alpha}} \ \mathcal{L}_{val}(\boldsymbol{\omega}^*(\boldsymbol{\alpha}), \boldsymbol{\alpha}) \qquad \text{s.t. } \boldsymbol{\omega}^*(\boldsymbol{\alpha}) = \mathrm{argmin}_{\boldsymbol{\omega}} \ \mathcal{L}_{train}(\boldsymbol{\omega}, \boldsymbol{\alpha}). \tag{6}$$

Here $\mathcal{L}_{train}$ and $\mathcal{L}_{val}$ denote the training and validation loss, respectively. $\boldsymbol{\alpha}$ is the architecture parameters calculating path strength. $\boldsymbol{\omega}$ is the network weights in MLP. Following the NAS-related works in the computer vision field (Liu et al., 2019b; Xie et al., 2019; Yao et al., 2020), we address this issue by first-order approximation. Specifically, we alternatively freeze architecture parameters $\boldsymbol{\alpha}$ when training $\boldsymbol{\omega}$ on the training set and freeze $\boldsymbol{\omega}$ when training $\boldsymbol{\alpha}$ on the validation set.

However, a recent research (Wang et al., 2021) has verified that architecture parameters are not effective enough to indicate the strength of the candidate operations. To search more effective meta-paths, different from previous methods (Liu et al., 2019a; Guo et al., 2020; Wang et al., 2021), we use

Table 1: Time complexity comparison of every training epoch. † means time complexity under small-scale datasets and full-batch training.

| Method | Feature projection | Neighbor aggregation | Semantic fusion | Total |
|---|---|---|---|---|
| HAN | $O(N(rd)^l F^2)$ | $O(N(rd)^l F)$ | $O(Nr^l F^2)$ | $O(N(rd)^l F^2)$ |
| simple-HGN | $O(N(rd)^l F^2)$ | $O(N(rd)^l F)$ | - | $O(N(rd)^l F^2)$ |
| simple-HGN† | $O(NrdlF^2)$ | $O(NrdlF)$ | - | $O(NrdlF^2)$ |
| SeHGNN | $O(Nr^l F^2)$ | - | $O(Nr^l F^2 + Nr^{2l}F)$ | $O(N(r^l F^2 + r^{2l}F))$ |
| LMSPS-search | $O(NMF^2)$ | - | $O(NMF^2)$ | $O(NMF^2)$ |
| LMSPS-train | $O(NMF^2)$ | - | $O(NMF^2)$ | $O(NMF^2)$ |

the path strength determined by architecture parameters to progressively narrow the search space to a size of $C = 2M$ based on Equation 5. Then, using path strength as the probability, we sample $M$ meta-paths from the compact search space $\mathbb{S}_{2M}$ to evaluate their performance. This sampling process can be represented as:

$$\bar{\mathbb{S}} = \text{DiscreteSample}(\mathbb{S}_{2M}, M, \bar{\mathbb{Q}}). \tag{7}$$

Here, $\text{DiscreteSample}(\mathbb{S}_{2M}, M, \bar{\mathbb{Q}})$ indicates a set of $M$ elements chosen from the set $\mathbb{S}_{2M}$ without replacement via discrete probability distribution $\bar{\mathbb{Q}}$. $\bar{\mathbb{Q}}$ is the set of relative path strength calculated by architecture parameters of the $2M$ meta-paths based on Equation 2. The sampling evaluation is repeated 200 times to filter out the meta-path set with the lowest validation loss. This stage is not time-consuming because the evaluation does not involve weight training. The overall algorithm is shown in the Appendix.

After finishing the search stage, the retained meta-path set is $\mathbb{S}_M$, containing the indexes of top-$M$ meta-paths. The forward propagation of the target-net can be formulated as:

$$\hat{Y} = \text{MLP}\Big( \big\|_{k \in \mathbb{S}_M} \text{MLP}_k(\boldsymbol{X}_k) \Big). \tag{8}$$

Here $\big\|$ denotes the concatenation operation. Unlike existing HGNNs like SeHGNN, the architecture of the target-net does not contain neighbor attention and semantic attention. Instead, the parametric modules consist of pure MLPs.

## 5.2 Time Complexity Analysis

Following the convention (Chiang et al., 2019; Yang et al., 2023), we compare the time complexity of LMSPS with HAN (Wang et al., 2019), Simple-HGN (Lv et al., 2021), and SeHGNN (Yang et al., 2023) under mini-batch training with the total $N$ target training nodes. All methods employ $l$-hop neighborhood. For simplicity, we assume that the number of features is fixed to $F$ for all layers. The average degree of each node is $rd$, where $r$ is the number of node-related edge types and $d$ is the number of edges connected to the node for each edge type.

The complexity analysis is summarized in Table 1. Because HAN and Simple-HGN require neighbor aggregation during training, and the number of neighbors grows exponentially with hops. they have neighbor aggregation costs of $O((rd)^l)$. SeHGNN employs an pre-processing step to avoid the training cost of neighbor aggregation. However, the number of meta-paths grows exponentially with hops, causing SeHGNN to suffer from $O(r^l)$ costs in semantic aggregation. Unlike the exponential methods above, LMSPS samples $M$ meta-paths in each iteration of the search stage and employs $M$ meta-paths in the training stage to avoid exponential growth. Generally, we have $O((rd)^l) \gg O(r^l) = O(K) \gg O(M)$ when $l$ is large. It is worth noting that the time complexity of LMSPS does not increase with maximum hop $l$, which is the key point for utilizing long-range meta-paths.

## 6 Experiments and Analysis

In this section, we evaluate the benefits of our method against state-of-the-art models on eight heterogeneous datasets. We aim to answer the following questions: **Q1.** How does LMSPS perform on large-scale and small-scale datasets compared with state-of-the-art baselines? **Q2.** Does LMSPS perform better on sparser large-scale HINs? **Q3.** Are the search algorithm and searched meta-paths effective? **Q4.** How does LMSPS perform when the value of maximum hop increases? More ablation studies and hyper-parameter studies are provided in the Appendix.

Table 2: Test accuracy (%) on the ogbn-mag compared with other methods on the OGB leaderboard. See main text for details of *label* and *ms*.

| Method | Test accuracy |
|---|---|
| *Random* | 35.14±3.78 |
| GraphSAGE (Hamilton et al., 2017) | 46.78±0.67 |
| RGCN (Schlichtkrull et al., 2018) | 47.37±0.48 |
| HGT (Hu et al., 2020) | 49.27±0.61 |
| NARS (Yu et al., 2020) | 50.88±0.12 |
| SeHGNN (Yang et al., 2023) | 51.79±0.24 |
| LMSPS | **53.56±0.19** |
| SAGN (Sun et al., 2021)+*label* | 51.17±0.32 |
| GAMLP (Zhang et al., 2022)+*label* | 51.63±0.22 |
| SeHGNN+*label* | 53.99±0.18 |
| LMSPS+*label* | **55.54±0.21** |
| SeHGNN+*label*+*ms* | 56.71±0.14 |
| LMSPS+*label*+*ms* | **57.67±0.15** |

Table 3: Micro-F1 (%) of LMSPS and baselines on the three datasets, *i.e.*, DBLP, IMDB, and ACM, from HGB benchmark. All methods do not employ data enhancements like label aggregation.

| Method | DBLP | IMDB | ACM |
|---|---|---|---|
| RSHN (Zhu et al., 2019) | 93.81±0.55 | 64.22±1.03 | 90.32±1.54 |
| HetSANN (Hong et al., 2020) | 80.56±1.50 | 57.68±0.44 | 89.91±0.37 |
| GTN (Yun et al., 2019) | 93.97±0.54 | 65.14±0.45 | 91.20±0.71 |
| HGT (Hu et al., 2020) | 93.49±0.25 | 67.20±0.57 | 91.00±0.76 |
| Simple-HGN (Lv et al., 2021) | 94.46±0.22 | 67.36±0.57 | 93.35±0.45 |
| HINormer (Mao et al., 2023) | 94.94±0.21 | 67.83±0.34 | - |
| RGCN (Schlichtkrull et al., 2018) | 92.07±0.50 | 62.05±0.15 | 91.41±0.75 |
| HetGNN (Zhang et al., 2019) | 92.33±0.41 | 51.16±0.65 | 86.05±0.25 |
| HAN (Wang et al., 2019) | 92.05±0.62 | 64.63±0.58 | 90.79±0.43 |
| MAGNN (Fu et al., 2020) | 93.76±0.45 | 64.67±1.67 | 90.77±0.65 |
| SeHGNN (Yang et al., 2023) | 95.24±0.13 | 68.21±0.32 | 93.87±0.50 |
| DiffMG (Ding et al., 2021) | 94.20±0.36 | 59.75±1.23 | 88.07±3.04 |
| PMMM (Li et al., 2023) | 95.14±0.22 | 67.58±0.22 | 93.71±0.17 |
| LMSPS | **95.59±0.16** | **68.82±0.19** | **94.56±0.17** |

## 6.1 DATASETS AND BASELINES

We evaluate LMSPS on the large-scale dataset ogbn-mag from OGB challenge (Hu et al., 2021), and three widely-used HINs including DBLP, IMDB, and ACM from HGB benchmark (Lv et al., 2021) on the node classification task. Please see Appendix A.1 for details of all datasets. For the ogbn-mag dataset, the results are compared to the OGB leaderboard, and all scores are the average of 10 separate training. For the three datasets from HGB, the results are compared to the baseline scores from the HGB paper and several recent advanced methods. Following the convention (Lv et al., 2021; Yang et al., 2023), all scores are the average of 5 separate local data partitions. GEMS (Han et al., 2020) and RMS-HRec (Ning et al., 2022) are ignored because they are designed for recommendation tasks. RL-HGNN (Zhong et al., 2020) is omitted due to a lack of source code.

## 6.2 EXPERIMENTAL SETUP

We set the number of selected meta-paths $M = 30$ for all datasets. The maximum hop is 6 for ogbn-mag, DBLP and 5 for IMDB, ACM. A two-layer MLP is adopted for each meta-path in the feature projection step and the hidden size is 512. $\tau$ linearly decays with the number of epochs from 8 to 4. All architecture parameters are initialized as 1s. Network weights are initialized by the Xavier uniform distribution (Glorot & Bengio, 2010) and are optimized with Adam (Kingma & Ba, 2015) during training. The learning rate is 0.001 and the weight decay is 0. For searching in the super-net, we train for 200 epochs with $\lambda = 1$ during the first 20 epochs for warmup and decreasing to 0 linearly. For training of the target-net, we use an early stop mechanism based on the validation performance to promise full training. All the experiments are conducted on a Tesla V100 16GB GPU.

## 6.3 PERFORMANCE COMPARISON

**Results on Ogbn-mag.** To answer **Q1**, we report the test accuracy of our proposed LMSPS and baselines in Table 2. *label* indicates a kind of data augmentation utilizing labels as extra inputs to provide enhancements (Wang & Leskovec, 2020; Shi et al., 2021; Yang et al., 2023). *ms* means using multi-stage learning (Li et al., 2018; Sun et al., 2020). We report results without or with *label* and *ms* for a comprehensive comparison. *Random* means the result of replacing our searched meta-paths with 30 random meta-paths. We show the average result of 20 random samples. It can be seen that LMSPS achieves state-of-the-art performance under each condition. For example, LMSPS outperforms the SOTA method SeHGNN by a large margin of 1.77%. The advantage of LMSPS compared to SeHGNN mainly comes from the ability to discover effective long-range meta-paths automatically. SeHGNN shows the results with extra embeddings as an enhancement in the original paper. Though we cannot compare results with this trick due to the untouchability of its embedding file, LMSPS still outperforms their best result.

**Performance on HGB Benchmark.** Table 3 shows the Micro-F1 scores of LMSPS and baselines. The 1st, 2nd, and 3rd blocks are metapath-free, metapath-based, and NAS-based methods, respectively.

Table 4: Results of LMSPS and SeHGNN on the sparse large-scale HINs. ↑ means the improvements in test accuracy.

| Dataset | SeHGNN | LMSPS | ↑ |
|---|---|---|---|
| ogbn-mag-5 | 36.04±0.64 | 40.13±0.34 | **4.09** |
| ogbn-mag-10 | 38.27±0.19 | 41.18±0.18 | **2.91** |
| ogbn-mag-20 | 39.18±0.09 | 41.40±0.13 | **2.22** |
| ogbn-mag-50 | 39.50±0.13 | 41.32±0.14 | **1.82** |

Table 5: Experiments on the generalization of the searched meta paths. * means using the meta-paths searched in LMSPS.

| Method | DBLP | IMDB | ACM |
|---|---|---|---|
| HAN | 92.05±0.62 | 64.63±0.58 | 90.79±0.43 |
| HAN* | 93.54±0.15 | 65.89±0.52 | 92.28±0.47 |
| SeHGNN | 95.24±0.13 | 68.21±0.32 | 93.87±0.50 |
| SeHGNN* | 95.57±0.23 | 68.59±0.24 | 94.46±0.18 |

The 4th block is our method. Every block contains a method yielding highly competitive performance, indicating that the HGNN field is very active and each mechanism has its advantage. However, LMSPS can still consistently achieves the best performance on all the three datasets.

## 6.4 The Necessity of Long-range Dependency

To answer **Q2** and explore the necessity of long-range dependency on HINs, we construct four large-scale datasets with high sparsity based on `ogbn-mag`. To avoid inappropriate preference seed settings of randomly removing, we construct fixed HINs by limiting the maximum in-degree related to edge type. Specifically, we gradually reduce the maximum in-degree related to edge type in `ogbn-mag` from 50 to 5 but leave all nodes unchanged. Details of the four datasets are listed in Appendix A.1. The test accuracy of LMSPS and SOTA method SeHGNN are shown in Table 4. LMSPS outperforms SeHGNN more significantly with the increasing sparsity. In addition, the leading gap of LMSPS over SeHGNN is more than $4\%$ on the highly sparse dataset `ogbn-mag-5`. The main difference between SeHGNN and LMSPS is that the former cannot utilize large hops and only use hop 2 while the latter has a maximum hop of 6, demonstrating that long-range dependencies are more effective with increased sparsity and decreased direct neighbors in HINs.

## 6.5 Effectiveness of the Searched Meta-paths

To demonstrate the effectiveness of searched meta-paths, on one hand, the meta-paths should be effective in the proposed model, which the above experiments have verified. On the other hand, the meta-paths should be effective after being generalized to other HGNNs, which is important but has been ignored by previous works. To answer **Q3**, we verify the generalization of our searched meta-path on the most famous HGNN HAN (Wang et al., 2019) and the SOTA method SeHGNN (Yang et al., 2023). The Micro-F1 scores on three representative datasets are shown in Table 5. After simply replacing the original meta-path set with our searched meta-paths and keeping other settings unchanged, the performance of both methods improves rather than decreases, demonstrating the effectiveness of our searched meta-paths. We also explore the effectiveness of our search algorithm. In our architecture, our meta-paths are replaced by those meta-paths discovered by other methods. DARTS (Liu et al., 2019a) is the first differentiable search algorithm in neural networks. SPOS (Guo et al., 2020) is a classic singe-path differentiable algorithm. DiffMG (Ding et al., 2021) and PMMM (Li et al., 2023) search for meta-graphs instead of meta-paths. The derivation strategies of the four methods are unsuitable for discovering multiple meta-paths. So we changed their derivation strategies to ours to improve their performance. We report the Micro-F1 scores in Table 6. It can be seen that LMSPS shows the best performance, verifying the effectiveness of our search algorithm.

## 6.6 Analysis on Large Maximum Hops

To answer **Q4**, we compare the performance and training time of LMSPS with SeHGNN under different maximum hops on large-scale dataset `ogbn-mag`. When the maximum hop $l = 1, 2, 3$, we utilize the full meta-paths set because the number of target-node-related meta-paths is smaller than $M = 30$. Following the convention (Lv et al., 2021; Yang et al., 2023), we measure the average time consumption of one epoch for each model. As shown in Table 7, both the performance of LMSPS and SeHGNN increases with the growth of maximum hop value. Moreover, when the number of target-node-related meta-paths is larger than 30, the training time of LMSPS is stable under different maximum hops. We also conduct experiments to compare the performance, memory, and efficiency of LMSPS with the best metapath-free method with source code, Simple-HGN (Lv

Table 6: Experiments on the effectiveness of our search algorithm. In our LMSPS, the meta-paths are replaced by those discovered by other methods.

| Method | DBLP | IMDB | ACM |
|---|---|---|---|
| HAN | 95.44±0.14 | 65.95±0.31 | 90.66±0.30 |
| GTN | 95.33±0.05 | 65.99±0.16 | 90.66±0.30 |
| DARTS | 95.35±0.17 | 66.23±0.14 | 93.45±0.13 |
| SPOS | 95.41±0.43 | 67.10±0.29 | 93.64±0.37 |
| DiffMG | 95.45±0.49 | 66.98±0.37 | 93.61±0.45 |
| PMMM | 95.48±0.27 | 67.49±0.24 | 93.74±0.22 |
| LMSPS | 95.59±0.16 | 68.82±0.19 | 94.56±0.17 |

Table 7: Experiments on ogbn-mag to analyze the performance of SeHGNN and LMSPS under different maximum hops. #MP is the number of target-node-related meta-paths. "OOM" means out of memory.

| Max hop | #MP | SeHGNN | | LMSPS | |
|---|---|---|---|---|---|
| | | Time | Test accuracy | Time | Test accuracy |
| 1 | 4 | 4.35 | 47.18±0.28 | 3.98 | 46.88±0.10 |
| 2 | 10 | 6.44 | 51.79±0.24 | 5.63 | 51.91±0.13 |
| 3 | 23 | 11.28 | 52.44±0.16 | 10.02 | 52.72±0.24 |
| 4 | 50 | OOM | OOM | 14.34 | 53.33±0.18 |
| 5 | 107 | OOM | OOM | 14.77 | 53.42±0.21 |
| 6 | 226 | OOM | OOM | 14.71 | 53.56±0.19 |

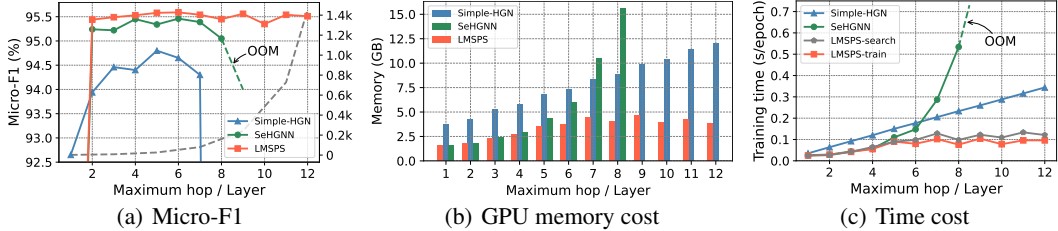

(a) Micro-F1      (b) GPU memory cost      (c) Time cost

Figure 3: Illustration of (a) performance, (b) memory cost, (c) average training time of Simple-HGN, SeHGNN, and LMSPS relative to the maximum hop or layer on `DBLP`. The gray dotted line in (a) indicates the number of target-node-related meta-paths under different maximum hops.

et al., 2021), and the best metapath-based method, SeHGNN (Yang et al., 2023), on `DBLP`. Figure 3 (a) shows that LMSPS has consistent performance with the increment of maximum hop. The failure of Simple-HGN demonstrates the attention mechanism can not eliminate the effects of noise under large hop. Figure 3 (b), (c) illustrate each training epoch's average memory and time costs relative to the maximum hop or layer. We can observe that the consumption of SeHGNN exponentially grows, and the consumption of Simple-HGN linearly increases, which is consistent with their time complexity as listed in Table 1. At the same time, LMSPS has almost constant consumption as the maximum hop grows. Figure 3 (c) shows the time cost of LMSPS in the search stage, which also approximates a constant when the number of meta-paths is larger than $M = 30$.

## 6.7 DISCUSSION

The performance of LMSPS does not always increase with the value of the maximum hop, and the best maximum hop depends on the dataset. For instance, LMSPS can effectively utilize 12-hop meta-paths on `DBLP` with high performance and low cost. However, `DBLP` is not very sparse, so LMSPS shows the best performance when the maximum hop is 6 because longer meta-paths may bring noise. In Table 4, we conduct experiments to demonstrate that longer meta-paths are more useful for sparser HINs. In addition, based on Table 1, the time complexity of LMSPS does not increase with the maximum hop. Consequently, it provides an effective solution for utilizing long-range dependency on HINs for possible applications. In future work, we would consider introducing residual mechanisms into the meta-path to boost the performance of longer meta-paths.

## 7 CONCLUSION

This work presented Long-range Meta-path Search through Progressive Sampling (LMSPS) to address the issue of leveraging long-range dependencies in heterogeneous information networks, including high costs and the difficulty in utilizing effective information. Based on the two findings on meta-paths, LMSPS introduced a progressive sampling search algorithm and a sampling evaluation strategy to automatically search effective meta-paths, thus reducing the exponentially increased number of meta-paths to a constant. Extensive experiments verified the superiority of LMSPS over state-of-the-art methods, especially on sparse heterogeneous graphs requiring long-range dependencies.

REPRODUCIBILITY STATEMENT

We have provided the motivation for dataset selection and listed hyper-parameters used for the experiments. All reported results are the average of multiple experiments with standard deviations. We have included a pseudocode description of our method in the Appendix. The source code has been submitted as supplementary materials with clear commands on reproducing our results.

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

# A APPENDIX

This supplementary material provides additional details and results not included in the main text due to space limitations. Concretely, we begin by displaying the details of all datasets and search objective in our experiments. Secondly, we show the search algorithm of LMSPS. Thirdly, we compare LMSPS with advanced baselines on training efficiency. Fourthly, we list and analyze the searched meta-paths and discuss over-smoothing and over-squashing issues. Fifthly, we compare LMSPS with SeHGNN on Freebase. Finally, we conduct ablation studies and hyperparameter studies.

Table 8: Statistics of datasets used in this paper.

| Dataset | #Nodes | #Node types | #Edges | #Edge types | Target | #Classes |
|---------|--------|-------------|--------|-------------|--------|----------|
| DBLP | 26,128 | 4 | 239,566 | 6 | author | 4 |
| IMDB | 21,420 | 4 | 86,642 | 6 | movie | 5 |
| ACM | 10,942 | 4 | 547,872 | 8 | paper | 3 |
| ogbn-mag | 1,939,743 | 4 | 21,111,007 | 4 | paper | 349 |
| ogbn-mag-50 | 1,939,743 | 4 | 9,531,403 | 4 | paper | 349 |
| ogbn-mag-20 | 1,939,743 | 4 | 7,975,003 | 4 | paper | 349 |
| ogbn-mag-10 | 1,939,743 | 4 | 6,610,464 | 4 | paper | 349 |
| ogbn-mag-5 | 1,939,743 | 4 | 4,958,941 | 4 | paper | 349 |

## A.1 MORE DATASET DETAILS

We evaluate our method on the large-scale dataset `ogbn-mag` from OGB challenge (Hu et al., 2021), and three widely-used HINs including `DBLP`, `IMDB`, and `ACM` from HGB benchmark (Lv et al., 2021). For the `ogbn-mag` dataset, we use the official data partition, where papers published before 2018, in 2018, and since 2019 are nodes for training, validation, and testing, respectively. The three datasets from HGB follow a transductive setting, where all edges are available during training, and target type nodes are divided into 24% for training, 6% for validation, and 70% for testing. In addition, we construct four sparse datasets to evaluate the performance of LMSPS by reducing the maximum in-degree related to edge type in `ogbn-mag`. The four sparse datasets have the same number of nodes with `ogbn-mag`. The details of all datasets are listed in Table 8.

## A.2 SEARCH OBJECTIVE

Following the convention(Lv et al., 2021; Li et al., 2023; Yang et al., 2023), we focus on semi-supervised node classification under the transductive setting. We use cross-entropy loss over all labeled nodes as:

$$\mathcal{L} = -\sum_{v \in \mathcal{V}_L} \sum_{n=1}^{N} \boldsymbol{y}_v[n] \log \bar{\boldsymbol{y}}_v[n],$$

where $N$ is the number of classes. $\mathcal{V}_L$ denotes the set of labeled nodes, $\boldsymbol{y}_v$ is a one-hot vector indicating the label of node $v$, and $\bar{\boldsymbol{y}}_v$ is the predicted label for the corresponding node in $\boldsymbol{Y}$ or $\hat{\boldsymbol{Y}}$. In the search stage, $\mathcal{V}_L$ is the training set when updating $\boldsymbol{\omega}$ and the validation set when updating $\boldsymbol{\alpha}$.

## A.3 ALGORITHM

Our search stage aims to discover the most effective meta-path set from all target-node-related meta-paths, severely challenging the efficiency of searching. Take ogbn-mag as an example, the number of target-node-related meta-paths is 226 and we need to find the most effective meta-path set with size 30. Because different meta-paths could be noisy or redundant to each other, top-30 meta-paths are not necessarily the optimal solution when their importance is calculated independently. Based on this consideration, the total number of meta-path sets is $C_{226}^{30} \approx 10^{37}$. Such a large search space is hard to solve efficiently by traditional RL-based algorithms (Zhong et al., 2020; Ning et al., 2022). To overcome this challenge, our LMSPS first uses a progressive sample algorithm to shrink the search space size from 226 to 60, then utilizes a sampling evaluation strategy to discover the best meta-path set.

---

**Algorithm 1** The search algorithm of LMSPS

---

**Input**: meta-path sets $\mathcal{P} = \{P_1, \cdots, P_K\}$; number of sampling meta-paths $M$; number of training iterations $T$; number of sampling evaluation $E$;
**Parameter**: Network weights $\boldsymbol{\omega}$ in $\text{MLP}_k$ for feature projection and MLP for downstream tasks; architecture parameters $\boldsymbol{\alpha} = \{\alpha_1, \cdots, \alpha_K\}$
**Output**: The index set of selected meta-paths $\mathcal{S}_M$

1: **% Neighbor aggregation**
2: Calculate neighbor aggregation of raw features for each $P_k \in \mathcal{P}$ based on Equation 1
3: **while** $t<T$ **do**
4:    **% Path Strength**
5:    Calculate the path strength of all meta-paths based on Equation 2
6:    **% Dynamic search space**
7:    Calculate the current search space $\mathcal{S}_C$ based on Equation 3 and Equation 4
8:    **% Sampling**
9:    Determine the indexes of sampled meta-paths based on Equation 5
10:    **% Semantic fusion**
11:    Fused the semantic information of the sampled meta-paths based on Equation 5
12:    **% Parameters updation**
13:    Update weights $\boldsymbol{\omega}$ by $\nabla_{\omega}\mathcal{L}_{train}(\boldsymbol{\omega}, \boldsymbol{\alpha})$
14:    Update parameters $\boldsymbol{\alpha}$ by $\nabla_{\alpha}\mathcal{L}_{val}(\boldsymbol{\omega}, \boldsymbol{\alpha})$
15: **end while**
16: **% Evaluation**
17: **while** $e<E$ **do**
18:    Randomly sample $M$ meta-paths from $\mathcal{S}_C$ as $\bar{\mathcal{S}}$ based on Equation 7
19:    Calculate $\mathcal{L}_{val}(\bar{\mathcal{S}})$ of the sampled meta-paths
20: **end while**
21: Select the best meta-path set for $\mathcal{S}_M \leftarrow \operatorname{argmin}_{\bar{\mathcal{S}}} \mathcal{L}_{val}(\bar{\mathcal{S}})$
22: **return** $\mathcal{S}_M$

---

We have introduced the components of LMSPS in detail in the main text. Generally, $K \gg M$ under a large maximum hop. For a small maximum hop, when $K \leq M$, the search stage is unnecessary because we can directly use all target-node-related meta-paths. When $K \leq 2M$, the progressive sampling algorithm is unnecessary because the search space is small enough. When $K > 2M$, we show the overall search algorithm in Algorithm 1.

### A.4  COMPARISON ON TRAINING EFFICIENCY

We show the time cost and parameters of LMSPS and the advanced baselines on `DBLP` and `ACM` in Figure 4. Following the convention (Lv et al., 2021; Yang et al., 2023), we measure the average time consumption of one epoch for each model. The area of the circles represents the parameters. The hidden size is set to $512$ and the maximum hop or layer is 6 for `DBLP` and 5 for `ACM` for all methods to test the training time and parameters under the same setting. Some methods perform quite poorly under the large maximum hop or layer. So we show the performance from Table 3 of the main text, which is the results under their best settings. Figure 4 shows that LMSPS has advantages in both training efficiency and performance. Our searched meta-paths are universal after searching once and can be applied to other meta-path-based HGNNs. We exclude our search time in Figure 4 because other meta-path-based HGNNs don't include the time for discovering manual meta-paths. The search time is shown in Figure 3 (c) of the main text.

### A.5  INTERPRETABILITY OF SEARCHED META-PATHS

We have conducted an extensive experimental study to validate the effectiveness of our searched meta-paths in the main text. Here, we illustrate the searched meta-paths of each dataset in Table 9. Because we discover many more meta-paths than traditional methods and most meta-paths are longer than traditional meta-paths, it is tough to interpret them one by one. So, we focus on the interpretability of meta-paths on `obgn-mag` from the Open Graph Benchmark. The `ogbn-mag` dataset is a heterogeneous network composed of a subset of the Microsoft Academic Graph. It

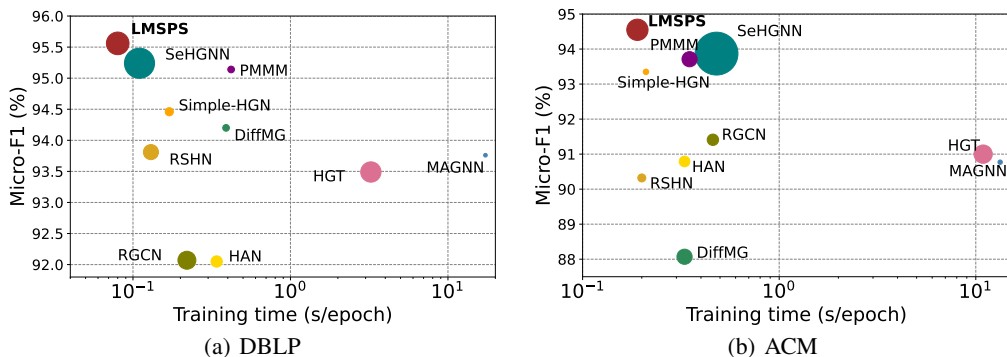

(a) DBLP  (b) ACM

Figure 4: Micro-F1 scores, time consumption, and parameters of various HGNNs on `DBLP` and `ACM`. GTN has a large time consumption and parameters. We ignore it for ease of illustration.

Table 9: Meta-paths searched by LMSPS on different datasets.

| Dataset | Meta-paths learnt by LMSPS |
|---|---|
| DBLP | AP, APT, APVP, APAPA, APTPA, APTPT, APTPV, APVPA, APVPV, APAPAP, APAPTP, APAPVP, APTPTP, APTPVP, APVPTP, APAPAPV, APAPTPA, APAPTPV, APAPVPA, APAPVPT, APAPVPV, APTPAPA, APTPAPT, APTPTPT, APTPVPV, APVPAPT, APVPTPA, APVPVPA, APVPVPT, APVPVPV |
| IMDB | M, MA, MK, MAM, MDM, MKM, MAMK, MDMK, MKMD, MDMKM, MKMAM, MKMDM, MKMKM, MAMAMK, MAMDMA, MAMDMK, MAMKMD, MAMKMK, MDMAMA, MDMAMD, MDMDMA, MDMDMK, MDMKMA, MKMAMA, MKMAMD, MKMAMK, MKMDMA, MKMDMK, MKMKMD, MKMKMK |
| ACM | PPP, PAPP, PCPA, PCPP, PPPC, PPPP, PAPAP, PAPPP, PCPAP, PCPPA, PPAPA, PPAPC, PPAPP, PAPAPA, PAPCPA, PAPPAP, PAPPCP, PAPPPP, PCPAPP, PCPCPP, PCPPAP, PCPPPP, PPAPAP, PPAPCP, PPAPPA, PPAPPP, PPCPAP, PPPAPA, PPPCPA, PPPPPP |
| ogbn-mag | PF, PAPF, PFPP, PAPPP, PFPFP, PPAPP, PPPAP, PPPFP, PAIAPP, PAPAPF, PFPAPF, PFPPPF, PFPPPP, PPAPPF, PPPAPF, PPPPAP, PPPPPF, PAIAPAP, PAPAPAP, PAPAPPP, PAPPPAI, PAPPPPF, PAPPPPP, PFPAPAP, PFPPFPF, PFPPPPF, PPAPAPF, PPAPAPP, PPPAIAI, PPPPAPPP |

includes four different entity types: papers (P), authors (A), institutions (I), and fields of study (F), as well as four different directed relation types: an author is "affiliated with" an institution, an author "writes" a paper, a paper "cites" a paper, and a paper "has a topic of" a field of study. The target node is the paper, and the task is to predict each paper's venue (conference or journal).

Based on Table 9, the hop of effective meta-paths on `obgn-mag` ranges from 1 to 6, which means utilizing information from neighbors at different distances is important. Because long-range meta-paths provide larger receptive fields, LMSPS shows stronger capability in message passing compared to traditional metapath-based HGNNs. The source node type of 16 meta-paths is P. It indicates that the neighborhood papers of the target paper are most significant for predicting its venue, which is consistent with reality: the citation relationship, coauthor relationship, and co-topic relationship between papers are usually the most effective information. 12 meta-paths' source node type is F. It implies that the neighborhood fields of the target paper are also crucial in determining its venue, which is also consistent with reality because most conferences or journals focus on a few fixed fields. The source node type of 2 meta-paths is I. It means the neighborhood institution is not very important for predicting the paper's venue, which is reasonable because almost all institutions have a wide range of conference or journal options. No meta-path has source node type A. It means the neighborhood author is unimportant in determining the paper's venue, which is logical because each paper has multiple authors, and each author can consider different venues. So, it is difficult to determine the paper's venue based on its neighborhood authors. If using too much institution or author information to predict the paper's venue, it actually introduces much useless information, which can be viewed as a kind of noise in `obgn-mag`.

Table 10: The comparison on Freebase. *label* indicates taking labels as extra inputs for enhancement.

| Methods | w/o *label* | | w *label* | |
|---------|----------|----------|----------|----------|
| | Macro-F1 | Micro-F1 | Macro-F1 | Micro-F1 |
| SeHGNN | 50.71±0.44 | 63.41±0.47 | 51.87±0.86 | 65.08±0.66 |
| LMSPS | 51.86±0.41 | 65.10±0.38 | 52.42±0.51 | 65.63±0.49 |

## A.6 DISCUSSION ON OVER-SMOOTHING AND OVER-SQUASHING

As far as we know, whether HGNNs also suffer from over-smoothing and over-squashing issues is still an open question. It can be discussed in two situations. HGNNs can be divided into two categories, i.e., metapath-free HGNNs and metapath-based HGNNs. The former (Zhu et al., 2019; Hong et al., 2020; Hu et al., 2020; Lv et al., 2021) directly incorporates the heterogeneity information in representation learning using a GNN-like version. So, these methods, e.g., HINormer (Mao et al., 2023), need to solve the over-smoothing and over-squashing issues. The latter (Ji et al., 2021; Yang et al., 2023; Wang et al., 2019; Fu et al., 2020) selectively aggregates messages with the assistance of manually designed or automatically generated meta-paths. The number of employed meta-paths typically does not increase with the hops, and each target node only incorporates a part of effective information. Consequently, these methods are typically not affected by over-smoothing and over-squashing issues.

In LMSPS, each target node only aggregates a part of neighborhood information with the guidance of effective meta-paths. It means the farther a node is from the target node, the less likely it is to be aggregated. So, each target node incorporates different information and the over-smoothing issue is at least alleviated. Additionally, we set the number of selected meta-paths $M = 30$ for all datasets. $M$ does not increase with maximum length/hops. So, the over-squashing issue does not exist.

## A.7 COMPARISON ON FREEBASE

Compared with `DBLP`, `IMDB`, `ACM`, and `ogbn-mag`, Freebase is less employed by related works. We ignore Freebase in the main text because of the space limitation. Here, we compare LMSPS with the SOTA method SeHGNN on Freebase in the Table 10 using the settings in our paper. As we can see, LMSPS shows better performance in both conditions.

## A.8 ABLATION STUDIES ON CRITICAL MODULES

Two components differentiate our LMSPS from other HGNNs: search algorithm and semantic fusion without attention. The search algorithm consists of a progressive sampling strategy and a sampling evaluation strategy. Additionally, most HGNNs employ the attention mechanism to fuse semantic information, increasing computational costs. In LMSPS, the difference of importance between the searched effective meta-paths is much smaller than that between the full meta-path set, making the attention mechanism unnecessary. We explore how each of them improves performance through ablation studies on `DBLP`. As shown in Table 11, the performance of LMSPS decreases when removing progressive sampling or sampling evaluation strategy. In addition, utilizing the full meta-path set decreases performance and increases training time, parameters, and memory costs, demonstrating the necessity of utilizing effective meta-paths. Transformers have shown great potential to become cross-field unified models. However, using transformers for semantic attention on searched meta-paths can not improve performance and harm efficiency, indicating semantic attention is not necessary for LMSPS. The reason behind this is the difference of importance between the searched effective meta-paths is much smaller than that between the full meta-path set in LMSPS.

## A.9 HYPERPARAMETER STUDY ON MAXIMUM HOP

To observe the impact of different maximum hop values, we show the Micro-F1 of LMSPS with respect to different maximum hop values on `DBLP`, `IMDB`, and `ACM` in Figure 5. We can see LMSPS show the best performance when the maximum hop is 5 or 6. Additionally, the performance of LMSPS does not always increase with the value of the maximum hop, the best maximum hop

Table 11: Experiments on DBLP to analyze the effects of different blocks in LMSPS. PS means progressive sampling strategy, and SE means sampling evaluation strategy.* means replacing searched meta-paths with the full meta-path set, and † means replacing the concatenation operation with the transformer module for semantic attention.

| Methods | Time/epoch | #Parameter | Memory | Macro-F1 | Micro-F1 |
|---|---|---|---|---|---|
| LMSPS | 0.08 | 43.6M | 3.8G | 95.32±0.19 | 95.59±0.16 |
| LMSPS w/o PS | 0.08 | 43.4M | 3.7G | 94.99±0.21 | 95.30±0.16 |
| LMSPS w/o SE | 0.09 | 44.6M | 4.1G | 95.19±0.18 | 95.48±0.14 |
| LMSPS* | 0.13 | 73.7M | 5.5G | 95.14±0.14 | 95.43±0.13 |
| LMSPS† | 0.10 | 45.0M | 4.2G | 95.24±0.25 | 95.57±0.23 |

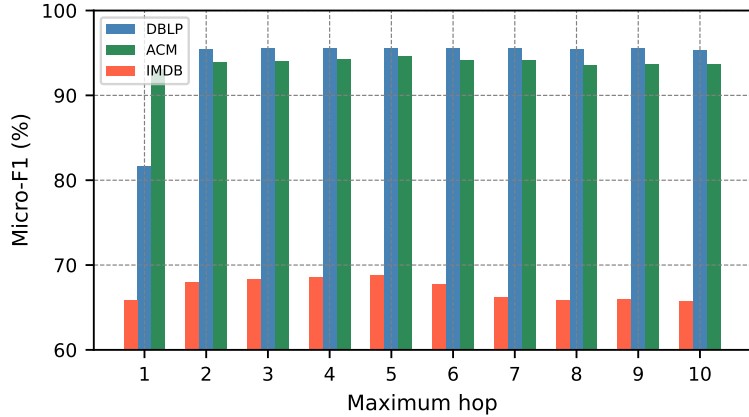

Figure 5: Micro-F1 scores with respect to different maximum hops on DBLP, IMDB, and ACM.

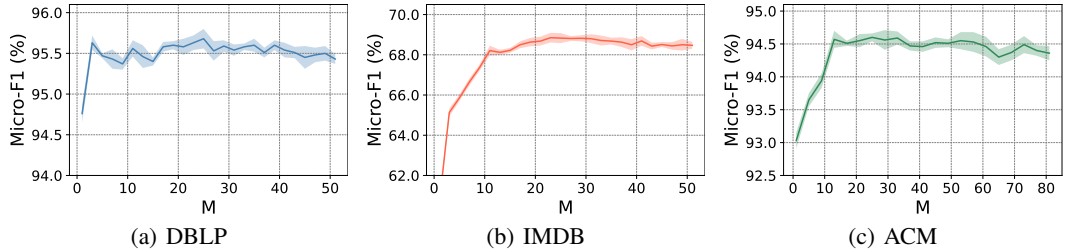

(a) DBLP     (b) IMDB     (c) ACM

Figure 6: Micro-F1 with respect to different hyper-parameter $M$ on DBLP, IMDB, and ACM.

depends on the dataset. It is reasonable because too long meta-paths may bring noise for a non-sparse dataset.

## A.10 HYPERPARAMETER STUDY ON $M$

In LMSPS, we randomly sampled $M$ meta-paths at each epoch in the search stage and selected the top-$M$ meta-paths in the training stage. Here, we perform analysis on hyper-parameter $M$ on DBLP, IMDB, and ACM. As illustrated in Figure 6, the performance of LMSPS increase with the growth of $M$ when $M$ is small. In addition, the performance decreases when $M$ is larger than a certain threshold, and the threshold varies with datasets. For unity, we set $M = 30$ for all datasets.

