# OpenReview forum: "Long-range Meta-path Search through Progressive Sampling on Large-scale Heterogeneous Information Networks"
_ICLR.cc/2024/Conference — Submitted to ICLR 2024_

### Official Review · Reviewer_f416 · 2023-10-25

**Soundness:** 3 good
**Presentation:** 3 good
**Contribution:** 2 fair
**Rating:** 5
**Confidence:** 4

**Summary:**

The paper addresses the challenge of utilizing long-range dependencies in large-scale heterogeneous information networks (HINs). The authors introduce an automatic framework named Long-range Meta-path Search through Progressive Sampling (LMSPS). This framework aims to discover meta-paths for various datasets or tasks without prior knowledge. The method involves developing a search space with all target-node-related meta-paths and then progressively shrinking this space. The approach is guided by a sampling evaluation strategy, leading to specialized and expressive meta-path selection. Experimental results on eight heterogeneous datasets show that LMSPS effectively discovers long-range meta-paths and outperforms existing models.

**Strengths:**

1 The idea of searching long-range meta-paths by developing a search space with all target-node-related meta-paths and then progressively shrinking this space is technical sound.

2 The paper is well written and organized

**Weaknesses:**

1 Why is the search for long-range meta-paths essential? Meta-paths are crucial because they convey rich semantic details. As these paths extend, it's uncertain whether they maintain their clear semantic significance.
2 Long-range meta-paths can be segmented into shorter meta-paths. If we can grasp these short-range meta-paths, then by using a compositional approach, we can understand long-range meta-paths.
3 Finding meta-structures that work effectively across various HGNNs is a tough task. It appears even more challenging to make long-range meta-path searches effective across different HGNNs. In contrast, short-range meta-paths might be more adaptable to various HGNNs.
4 The core idea revolves around creating a search space filled with all target-node-related meta-paths, which is then gradually reduced. Given that RL is seen as a potent tool for searching large spaces, how does the proposed method outperform other RL-based strategies?

**Questions:**

1 Why is the search for long-range meta-paths essential?
2 Why the proposed method is superior to the RL based methods.

---

> ### Author Response · Authors · 2023-11-21
> **Response to Reviewer f416 (Part 1)**
>
> Thanks for your constructive comments. In the following, we respond to your concerns point by point.
>
> ---
>
> **Point 1:** *Why is the search for long-range meta-paths essential?*
>
> **Reply 1:** As we stated in the fourth paragraph of Section 1, most metapath-based HGNNs, including famous HAN [a], MAGNN [b], and SeHGNN [c], obtain information from l-hop neighborhoods by utilizing meta-paths with the maximum hop $l$, i.e., all meta-paths are no more than $l$ hops/length.  So, long-range meta-paths mean larger receptive fields. Most metapath-based HGNNs integrate meta-path information in the first layer instead of inserting short meta-paths to each layer because multi-layer networks splicing semantics of different layers make high-level semantics indistinguishable. This phenomenon has been fully explored by SeHGNN. We put its main results below.
>
> Table 1: Experiments to analyze the effects of different combinations of the number of layers and the maximum meta-path hop. e.g., the structure (1,1,1,1) means a four-layer network with all meta-paths no more than 1 hop in each layer.
>
> | network      | DBLP (macro-f1) | DBLP (micro-f1) | ACM (macro-f1) | ACM (micro-f1) |
> | :----------- | :-------------: | :-------------: | :------------: | :------------: |
> | (1, 1, 1, 1) |      89.55      |      90.44      |     88.62      |     88.93      |
> | (2, 2)       |      91.88      |      92.35      |     92.57      |     92.53      |
> | (4)          |      93.60      |      94.02      |     92.82      |     92.79      |
>
>
> Therefore, long-range meta-paths are essential because they provide larger receptive fields and yield better performance than stacked short meta-paths. As shown in Table 9 in the Appendix, our method discovers short meta-paths as well as long meta-paths so as to effectively utilize information from neighbors at different distances. We have clarified it in Sections 1 and A.5 of the revised manuscript. Thank you!
>
> ---
>
> **Point 2:** *Long-range meta-paths can be segmented into shorter meta-paths.  If we can grasp these short-range meta-paths, then by using a compositional approach, we can understand long-range meta-paths.*
>
> **Reply 2:** Actually, long-range meta-paths is not harder to be understood than stacked short-range meta-paths. For example, we can easily distinguish high-level semantics such as being written by familiar authors (PAPAP). PAPAP can also be split into (PAP, PAP) or (PA, APA, AP) or (PA, AP, PA, AP). Understanding these short meta-paths and how they combine is not easier than directly understanding long ones. In addition, in Section A.5, we have shown the interpretability of the searched meta-path set on the large-scale dataset obgn-mag, which contains long-range meta-paths. Moreover, automatic meta-path search actually frees researchers from the understand-then-apply paradigm. The hand-crafted meta-paths rely on intense expert knowledge, which is both laborious and data-dependent. As shown in Table 9, our method can search effective meta-paths without prior knowledge for various datasets, which is even more valuable than existing several manually-defined meta-paths.
>
> ---
>
> **Point 3:** *Finding meta-structures that work effectively across various HGNNs is a tough task. Short-range meta-paths might be more adaptable to various HGNNs.*
>
> **Reply 3:** As we described in Reply 1, most metapath-based HGNNs integrate meta-path information in the first layer instead of inserting short meta-paths to each layer. So combined short-range meta-paths are hard to transform to most HGNNs due to structural conflict. Additionally, it is also hard to determine effective meta-paths in each layer and how to combine various meta-paths in different layers. At the same time, the combination depends on the network structure. So finding short-range meta-paths that work effectively across various HGNNs is even tougher. In contrast, the effective meta-paths mainly depend on the dataset instead of the architecture. As stated in Section 5.1, to avoid the influence of specific modules on search results, we utilize a simple MLP-based architecture to improve the generalizability of searched meta-paths. Therefore, our searched meta-paths can be transformed to HAN and SeHGNN and show great performance, as shown in Table 5.  We have clarified it more clearly in Section 1 of the revised manuscript. Thank you!

---

> ### Author Response · Authors · 2023-11-21
> **Response to Reviewer f416 (Part 2)**
>
> **Point 4:** *Why the proposed method is superior to the RL-based methods.*
>
> **Reply 4:** The main advantage is efficiency. As shown in Table 1 and Figure 3 (c) of the manuscript, our search cost is similar to training a single HGNN and the search cost almost does not increase with the maximum hop/length of meta-paths. RL is an important technology in neural architecture search (NAS). While RL shows promising performance in NAS, high cost is also one of its main features because it needs to train the model to converge for each candidate actions. For example, the famous NASNet [b] takes 2000 GPU hours to search for an effective architecture.
>
> Our search stage aims to discover the most effective meta-path set from all target-node-related meta-paths, severely challenging the efficiency of searching. Take ogbn-mag as an example, the number of target-node-related meta-paths is 226 and we need to find the most effective meta-path set with size 30. Because different meta-paths could be noisy or redundant to each other, top-30 meta-paths are not necessarily the optimal solution when their importance is calculated independently. Based on this consideration, the total number of meta-path sets is $C_{226}^{30} \approx 10^{37}$. Such a large search space on a large-scale dataset is hard to solve efficiently by RL-based methods, while LMSPS can finish searching in two hours. To achieve such high efficiency, our LMSPS first uses a progressive sample algorithm to shrink the search space size from 226 to 60, then utilizes a sampling evaluation strategy to discover the best meta-path set. We have clarified it in Section A.3 of the revised manuscript. Thank you!
>
> ---
>
>
> **References:**
>
> [a] [WWW 2019] Heterogeneous Graph Attention Network
>
> [b] [WWW 2020] MAGNN: Metapath Aggregated Graph Neural Network for Heterogeneous Graph Embedding
>
> [c] [AAAI 2023] Simple and Efficient Heterogeneous Graph Neural Network
>
> [d] [CVPR 2018] Learning Transferable Architectures for Scalable Image Recognition

---

> > ### Comment · Reviewer_f416 · 2023-11-22
> > **Response to the authors**
> >
> > I appreciate the thorough responses. My opinion of the paper remains unchanged.

---

### Official Review · Reviewer_vDix · 2023-10-30

**Soundness:** 2 fair
**Presentation:** 3 good
**Contribution:** 2 fair
**Rating:** 5
**Confidence:** 4

**Summary:**

In this paper, the authors focused on the graph representation learning task on heterogenous information networks. A novel LMSPS method was introduced to automatically select the proper set of long-range metapaths via a progressive sampling algorithm that dynamically shrink the search space with hop-independent time complexity. The overall presentation is clear. The experiments are also sufficient, where the proposed LMSPS method achieved impressive results. Moreover, the authors provided their source code for review.

**Strengths:**

S1. The overall presentation of this paper is clear, which is easy to grasp the key ideas.

S2. Using progressive sampling algorithm to search for effective meta-paths seems interesting.

S3. The proposed method achieved impressive results in the experiments.

S4. The authors provided their source code for review.

**Weaknesses:**

**W1. Some of the statements in this paper seems to be inconsistent or need further clarification.**

In Section 1, the authors argued that 'the receptive field grows exponentially with the number of layers' (for metapath-free HGNNs). Moreover, the authors also argue that many GNNs (e.g., for homogeneous graphs) tried to explore long-range dependency and gained some benefits. From my perspective, the step/range of a path is equivalent to the number of GNN layers in some cases, which indicates that the exploration of long-range dependency may still cause the issue of exponentially grown receptive field. As I known, many real-world graphs follow the property of 'six degrees of separation' (i.e., the average diameter of many real-world graphs would not be very large). In particular, when the path length and number of GNN layers are large (e.g., >>6), all the nodes may be in the receptive field centered at each node.

The availability of graph attributes (e.g., in terms of node attributes or edge attributes) is not mentioned in the problem statements of Section 2. However, as stated in the 2nd paragraph of Section 5.1, each node is associated with raw features. Details regarding graph attributes are also not described in Table 8.

In Section 3, the authors argued that existing methods (e.g., GTN, HGT, HAN, HPN, MEGNN, GraphMSE, etc.) automatically select a proper set of metapaths but are not as effective as the full meta-path set. However, according to my understanding, the proposed method still does not use the full meta-path set.

In Eq. (2), I am still confused about how to derive $\{ \alpha_k \}$. Are they model parameters to be learned or they are derived based on $\{ {\bf{X'}}_k \}$? If they are learnable parameters, it seems that the corresponding scale is related to $K$, which may still grow exponentially with the increase of path length.

The training loss shown in Eq. (6) includes the training and validations sets, which seems to be different from the standard supervised learning paradigms. It is suggested to highlight such a new paradigm (e.g., using formal math notations) at the very beginning of problem statements (e.g., in Section 2).

What are the training losses for 'LMSPS' and 'LMSPS+label' in Table 2? Does 'LMSPS' mean that it is trained with an unsupervised/self-supervised loss? If so, what is the definition of this loss? According to my understanding, the proposed method is supervised, which relies on the supervised loss as illustrated in Eq. (6). If so can the proposed method be extended to the unsupervised/self-supervised paradigm, where we could only train the model based on the original graph topology and attributes without any label information?

Why there are no results for 'LMSPS+label' and 'LMSPS+label+ms' in Table 3?

***

**W2. According to my understanding, the authors only tested the proposed method using the transductive setting of node classification. Its ability to handle the advanced inductive inference (e.g., for new unseen nodes and across graphs) was not validated in experiments.**

***

**W3. Some of the statements in the paper breaks the anonymity of this submission during the review period.**

The authors claimed that their method ranks 1st on the leaderboard of ogbn-mag in OGB (as shown in https://ogb.stanford.edu/docs/leader_nodeprop/). However, by checking this web page, I now clearly know the names and institutions of all the authors, which breaks the anonymity of this submission.

**Questions:**

See W1-W2

---

> ### Author Response · Authors · 2023-11-21
> **Response to Reviewer vDix (Part 1)**
>
> Thanks for your constructive comments. In the following, we respond to your concerns point by point.
>
> ---
>
> **W1-Point 1:** *The author ignores that long-range dependency may still cause the issue of an exponentially grown receptive field. In addition, over-smoothing is a problem.*
>
> **Reply 1:**  The reviewer might have overlooked the statement in the fourth paragraph of the Introduction section. For meta-path-based, the number of target-node-related meta-paths also grows exponentially with the maximum hop value increasing. I agree with your perspective that the maximum hop/step/range of meta-paths is equivalent to the number of HGNN layers and this perspective has been reflected in many places of our paper, e.g., the abscissa in Figure 3. **So our key contribution of reducing the exponentially increased number of meta-paths to a constant actually aims to solve the issue of exponentially grown receptive field.** We have clarified it more clearly in Section 1 of the revised manuscript.
>
> As far as we know, whether HGNNs also suffer from over-smoothing issues is still an open question. In our method, each target node only aggregates a part of neighborhood information with the guidance of effective meta-paths. It means the farther a node is from the target node, the less likely it is to be aggregated. So, each target node incorporates different information and the over-smoothing issue is at least alleviated. This conclusion is supported by Table 7, in which the performance of LMSPS on the representative ogbn-mag dataset keeps rising when maximum hop/length increases from $1$ to $6$. We have added a discussion on the over-smoothing issue in Section A.6 of the revised manuscript. Thanks.
>
> ---
>
> **W1-Point 2:** *The availability of graph attributes (e.g., in terms of node attributes or edge attributes) is not mentioned in the problem statements of Section 2. Details regarding graph attributes are also not described in Table 8.*
>
> **Reply 2:** Actually, all recent HGNNs do not describe the availability of node attributes or edge attributes in Preliminaries or list node attributes or edge attributes in Table, including the best metapath-free HGNN HINormer [a], the best metapath-based HGNN SeHGNN [b], and the best NAS-base HGNN PMMM [c]. The original paper of the HGB benchmark [d] also does not do so because the graph attributes can be easily accessed based on the corresponding citation. Following the convention, we introduce HINs and meta-paths in Preliminaries and list the node number, node types, edge number, edge types, target node type and classes of all datasets in Table 8.
>
> ---
>
> **W1-Point 3:** *In Section 3, the authors argued that existing methods select a proper set of meta-paths but are not as effective as the full meta-path set. However, according to my understanding, the proposed method still does not use the full meta-path set.*
>
> **Reply 3:** The reviewer might have some misunderstanding on our method. **One of our key contributions is the ability to utilize effective meta-paths instead of the full meta-path set.** If we use the full meta-path set, the computational cost is unacceptable under long-range dependency because the number of meta-paths grows exponentially with the maximum hop. Actually, in Section 3, our description is that some existing methods can implicitly learn meta-paths by attention, however, they use the full meta-path set instead of discovered meta-paths to generate the final results. Consequently, their learned meta-paths are not very effective and they can not expand to long-range dependency. We have clarified it more clearly in Section 3 of the revised manuscript.
>
> ---
>
> **W1-Point 4:** *If $\alpha_k$ are learnable parameters, it seems that the corresponding scale is related to $K$, which may still grow exponentially with the increase of path length.*
>
> **Reply 4:** Sorry for the confusion. $\alpha_k$ are learnable parameters. Based on Eq. (6), in the search stage, we freeze architecture parameters $\alpha$ when training $\omega$ on the training set and freeze $\omega$ when training $\alpha$ on the validation set. These two update processes are performed in an alternative manner. As stated in the fourth paragraph of Section 5.1, $\alpha_k$'s scale is related to $K$, severely challenging the efficiency of super-net training. Consequently, in each iteration, we only uniformly sample $M$ meta-paths from the whole search space for parameter updates, **so the search cost is relevant to $M$, which is a predefined small number, rather than $K$**. Because the search stage has many iterations and the initial values of architecture parameters are the same, all architecture parameters will be updated multiple times and the relative importance can be learned during training. So, as shown in Table 1, the time complexity of the search stage is related to $M$ and does not increase with the maximum hop/length. We have clarified it in Section 5.1 of the revised manuscript. Thank you!

---

> > ### Author Response · Authors · 2023-11-21
> > **Response to Reviewer vDix (Part 2)**
> >
> > **W1-Point 5:** *The training loss shown in Eq. (6) includes the training and validation sets, which seem to be different from the standard supervised learning paradigms.*
> >
> > **Reply 5:** Actually, Eq. (6) is one of the most famous equations and is widely used in neural architecture search (NAS). An approximate solution for this equation is to alternatively freeze architecture parameters $\alpha$ when training $\omega$ on the training set and freeze $\omega$ when training $\alpha$ on the validation set. Following the convention [c, e], we put Eq. (6) in Section 5.1 instead of the Preliminary section because it is difficult to understand when separated from the search algorithm. Note that the validation set is only employed in the search stage to determine effective meta-paths. In the re-training stage, we use standard semi-supervised paradigms. We have added Section A.2 in the revised manuscript to clarify it.  Thanks.
> >
> > ---
> >
> > **W1-Point 6:** *What are the training losses for 'LMSPS' and 'LMSPS+label'? Does 'LMSPS' mean that it is trained with an un/self-supervised loss? Can the method be extended to the un/self-supervised paradigm?*
> >
> > **Reply 6:** Sorry for the confusion. As stated in Section 6.3, $label$ means using labels as extra inputs to provide data enhancement. In this data enhancement, each node is prevented from receiving the ground truth label information of itself.  Following the convention [a,b,c,d],  we focus on semi-supervised node classification and the training loss of all our experiments is cross-entropy loss. So our model is not trained with an un/self-supervised loss. The method can not be extended to the un/self-supervised paradigm with its current version. However, our idea of using neural architecture search (NAS) to utilize long-range dependency can be expanded to the un/self-supervised paradigm by replacing the semi-supervised loss with un/self-supervised loss. We have clarified it more clearly in Section 6.3 of the revised manuscript. Thank you!
> >
> > ---
> >
> > **W1-Point 7:** *Why there are no results for 'LMSPS+label' and 'LMSPS+label+ms' in Table 3?*
> >
> > **Reply 7:** We do not use $label$ enhancement and $ms$ for fair comparison because all baselines did not employ the two enhancements in these three small datasets except SeHGNN. The ogbn-mag dataset brings an extra challenge that training nodes and test nodes are of different data distributions. Existing methods usually tackle these challenges by utilizing multi-stage learning (abbreviated as $ms$). Because our method aims to solve the challenges of utilizing long-range dependence in large-scale heterogeneous graphs and ogbn-mag is the most representative dataset, we report results without or with $label$ and $ms$ for a comprehensive comparison, as we stated in Section 6.3.
> >
> > ---
> >
> > **W2-Point 8:** *According to my understanding, the authors only tested the proposed method using the transductive setting of node classification.*
> >
> > **Reply 8:** Because heterogeneous graphs have more complex semantic information than homogeneous graphs and are much more challenging, as far as we know, most HGNNs focus on the transductive setting of node classification, including the best metapath-free HGNN HINormer [a], the best metapath-based HGNN SeHGNN [b], and the best NAS-base HGNN PMMM [c]. In addition, the most popular heterogeneous datasets for node classification, including ogbn-mag and the HGB Benchmark, are also under the transductive setting. As shown in Section 5, our method has great performance on all these datasets and shows high efficiency and generalization, which is even more powerful than existing SOTA models. The inductive setting on heterogeneous graphs is an interesting topic. We will try to explore it in future work. We have clarified it in Section A.2 of the revised manuscript. Thank you!
> >
> > ---
> >
> > **W3-Point 9:** *The authors claimed that their method ranks 1st on the leaderboard of ogbn-mag in OGB, which breaks the anonymity of this submission.*
> >
> > **Reply 9:** Based on the guidelines of the conference, it is OK to report the results on the leaderboard of a challenge based on the Author Guide (https://iclr.cc/Conferences/2024/AuthorGuide). We list the details below.
> >
> > **Q**: Can you explain how to treat de-anonymization in the case where a submitted paper refers to a challenge they won which can identify the authors?
> >
> > **A**: It is ok to report the results on the leaderboard of a challenge. The authors can include the ranking and the name of the challenge. The reviewers will be advised to not intentionally search the authors by examining the leaderboard.
> >
> > We feel this is similar to the arXiv submission, in that if the reviewer does not search on the web, it will be kept anonymous.
> >
> > ---
> >
> > We wish we had clarified all your confusion. If there are any further issues, please let us know, and we will be happy to continue to answer your questions. Thanks.

---

> ### Author Response · Authors · 2023-11-21
> **Response to Reviewer vDix (Part 3)**
>
> **References:**
>
> [a] [WWW 2023] HINormer: Representation Learning On Heterogeneous Information Networks with Graph Transformer
>
> [b] [AAAI 2023] Simple and Efficient Heterogeneous Graph Neural Network
>
> [c] [AAAI 2023] Differentiable Meta Multigraph Search with Partial Message Propagation on Heterogeneous Information Networks
>
> [d] [KDD 2021] Are we really making much progress? Revisiting, benchmarking, and refining heterogeneous graph neural networks
>
> [e] [KDD 2021] DiffMG: Differentiable Meta Graph Search for Heterogeneous Graph Neural Networks

---

### Official Review · Reviewer_dvsv · 2023-11-01

**Soundness:** 3 good
**Presentation:** 2 fair
**Contribution:** 2 fair
**Rating:** 5
**Confidence:** 3

**Summary:**

The paper proposes a novel approach called LMSPS for representation learning on heterogeneous information networks (HINs). LMSPS uses a progressive sampling algorithm to guide the selection of professional and expressive meta-paths, which capture the complex relationships between nodes in the HIN. The algorithm dynamically narrows down the search space by utilizing the characteristics of meta-paths, resulting in a compact search space that drives the current HIN and task. The authors demonstrate the effectiveness of LMSPS on several real-world datasets and show that it outperforms state-of-the-art methods in terms of both accuracy and efficiency. The main contribution of the paper is the development of a new approach for representation learning on HINs that achieves state-of-the-art performance while being computationally efficient.

**Strengths:**

(1) It propose a novel meta-path search framework for the first to utilize long-range dependency in large-scale HINs.

(2) It ranks top-1 on the leaderboards of ogbn-mag in Open Graph Benchmark to prove the feasibility of the method.

**Weaknesses:**

(1) The writing of this article is somewhat difficult to understand. For example, there is no specific introduction to the meaning of meta-path.

(2) In the fourth part, you used DBLP as an example to illustrate that the length of the meta-path affects the experimental results. Currently, most experiments set the meta-path length to 2, and it is believed that longer paths will affect the experiments. However, in Table 9, the length of the meta-path has been It can reach 6, which is contrary to the initial statement.

(3) Figure 2 does not match well with Part 5. For example, it does not reflect progressive sampling search and sampling evaluation. At the same time, the calculation method of the initial weight is not well reflected in the figure.

(4) It is not explained clearly why some meta-paths are used as noise. I hope I can write down the specific reasons.

**Questions:**

In Figure 1, I found that the results obtained by using all paths are actually similar to the results obtained by only APV. I don't quite understand the meaning of using meta paths? Would it be much different from your experiment to only use a path length of 2 or 3?

---

> ### Author Response · Authors · 2023-11-21
> **Response to Reviewer dvsv (Part 1)**
>
> Thanks for your constructive comments. In the following, we respond to your concerns point by point.
>
> ---
>
> **Point 1:** *The writing of this article is somewhat difficult to understand. For example, there is no specific introduction to the meaning of meta-path.*
>
> **Reply 1:** Sorry for the confusion. We have indeed used two paragraphs to introduce the definition and meaning of meta-path in Section 2. Here, we introduce the meta-path with the example in the Introduction section. The academic network, ogbn-mag, contains multiple node types, i.e., Paper (P), Author (A), Institution (I), and Field (F), as well as multiple edge types, such as Author $\xrightarrow{\rm writes}$ Paper, Paper$\xrightarrow{\rm cites}$Paper, Author $\xrightarrow{\rm affiliated}$ Institution, Paper$\xrightarrow{\rm topic}$Field. These elements can be combined to build higher-level semantic relations called meta-paths. For instance, PAP (Paper to Author to Paper) is a $2$-hop meta-path describing the co-author relationship, which corresponds to multiple meta-path instances (PAP paths) in the underlying ogbn-mag. In metapath-based HGNNs, each node uses the mean aggregator to aggregate features from the metapath-based neighbors for each given meta-path.
>
> ---
>
> **Point 2:** *In the fourth part, you used DBLP as an example to illustrate that the length of the meta-path affects the experimental results. Currently, most experiments set the meta-path length to 2. However, in Table 9, the length of the meta-path has been It can reach 6, which is contrary to the initial statement.*
>
> **Reply 2:** The reviewer might have misunderstood the fourth part and overlooked some details. Section 4 shows two exploratory experiments to introduce our motivation for meta-path search. We use DBLP to analyze the importance of different meta-paths one by one and obtain a finding that **a few meta-paths provide major contributions instead of the length of meta-paths affects the performance**. Then, we can employ a part of effective meta-paths instead of the full meta-path set without sacrificing performance when the maximum hop is large. As we stated in the second paragraph of Section 4, we set the maximum length $l = 2$ for ease of illustration, which is reasonable for an exploratory experiment. If we use too large maximum lengths in Section 4, the number of full meta-paths can reach hundreds. Then it will be hard for us to explore their performance one by one. In Section 6.2, we show the experimental setup for all datasets. **The maximum length is set to $6$ for DBLP**. Therefore, all results in the Experiment section use the same settings. We have tried to clarify it in Section 4 of the revision. Thanks.
>
> ---
>
> **Point 3:** *Figure 2 does not match well with Part 5. In addition, the calculation method of the initial weight is not well reflected in the figure.*
>
> **Reply 3:** Figure 2 shows the overall framework of LMSPS, including the pre-precessing, search, and training stages. The details of progressive sampling search and sampling evaluation are hard to illustrate in the limited space due to their complexity. For the calculation method of the initial weight, as we stated in Section 6.2, all the network parameters are initialized by the Xavier uniform distribution. All architecture parameters are initialized as $1$ to keep the same initial path strength. We have added more details in Figure 2 and Section 6.2 based on your suggestion. Thanks.
>
> ---
>
> **Point 4:** *It is not explained clearly why some meta-paths are used as noise. I hope I can write down the specific reasons.*
>
> **Reply 4:** Here, we still use ogbn-mag as an example, which includes four different entity types: papers (P), authors (A), institutions (I), and fields of study (F). The target node is the paper, and the task is to predict each paper's venue (conference or journal). As we state in Section A.5, i.e., interpretability of searched meta-paths, some meta-paths with source node type institutions (I) can be regarded as noise as it negatively affects the prediction accuracy. It is reasonable because almost all institutions have a wide range of conference or journal options. If we use the institution information to predict the paper's venue, e.g., PAPAI (P$\leftarrow$A$\leftarrow$P$\leftarrow$A$\leftarrow$I) meta-path, it actually introduces much useless information. In addition, as shown in Figure 1 (c), after removing meta-paths PC and PCP, the Micro-F1 score of SeHGNN on ACM improves by $0.52$%, indicating that PC and PCP are a kind of noise in the ACM dataset. We have clarified it more clearly in Section A.5 of the revised manuscript.

---

> > ### Author Response · Authors · 2023-11-21
> > **Response to Reviewer dvsv (Part 2)**
> >
> > **Point 5:** *I don't quite understand the meaning of using meta-paths. Would it be much different from your experiment to only use a path length of 2 or 3?*
> >
> > **Reply 5:** Because the number of meta-paths exponentially increases with the maximum length, we set the maximum length $l = 2$ for ease of illustration in Figure 1. There are only $5$ meta-paths on DBLP under the maximum length of 2. Therefore, Figure 1 (b) is a special case in that utilizing only one meta-path has a similar performance to using all meta-paths due to the small maximum length. As we explain in Reply 1, in metapath-based HGNNs, each node uses the mean aggregator to aggregate features from metapath-based neighbors for each related meta-path. So, **using meta-paths means selectively aggregating neighborhood information**. As shown in Table 7, both the performance of SeHGNN and our LMSPS on the representative large-scale dataset ogbn-mag increase with the maximum meta-path length. For example, the test accuracy of LMSPS increases from $51.91$% under maximum length $2$ to  $53.56$% under maximum length $6$. We have clarified it more clearly in Section 2 of the revised manuscript. Thanks.

---

### Official Review · Reviewer_kpcU · 2023-11-03

**Soundness:** 3 good
**Presentation:** 2 fair
**Contribution:** 3 good
**Rating:** 5
**Confidence:** 4

**Summary:**

The paper presents a study on the use of long-range dependencies in large-scale heterogeneous information networks (HINs). The authors address this by presenting an automatic framework named Long-range Meta-path Search through Progressive Sampling (LMSPS). LMSPS is designed to identify meta-paths within HINs effectively, without needing prior knowledge. It includes a search space that encompasses all meta-paths related to target nodes. Through a progressive sampling algorithm, the framework narrows this search space efficiently, achieving a reduced search space that is specific to the given HIN and task. A sampling evaluation strategy is used to select the most expressive and task-specific meta-paths. The results from experiments on eight different datasets indicate that LMSPS not only identifies effective long-range meta-paths but also surpasses existing models in performance.

**Strengths:**

1. The authors of the paper provided clear motivations for long-range metapath search. The experiments in Section 4 also provide useful insights into the problem.

2.  Experimental results not only show good performance, but also excellent efficiency: both GPU memory cost and computational is scalable.

3. The methodology introduced is both intuitive and technically sound, suggesting a well-reasoned approach.

**Weaknesses:**

W1. Long-range dependencies in graph neural networks are often associated with oversmoothing and over-squashing issues. While these issues are typically associated with the number of GCN layers, long meta-paths used may bring in similar problem. Some discussion in this aspect is needed.

W2. Transformers are becoming popular in graphs as well to capture long range dependencies, although they do suffer from scalability issues when the receptive field is too large. Hence, some methods like HINormer (Mao et al., 2023), resorts to subgraph based sampling. By improving the sampling strategy beyond just subgraphs, I think transformers are promising in addressing these issues. Some discussion on the advantages and disadvantages of transformers compared to this work should be elaborated.

W3. Writing is not very clear, especially in Section 5.1. It is more like a step-by-step introduction of the pipeling, lacking a clear, high-level organization so that readers know which are the important contributions and focuses.

Minor issue: In Figure 2, search stage, i'm not sure if the example 0/1 values given (in the green block) is right?

**Questions:**

Please see weaknesses.

---

> ### Author Response · Authors · 2023-11-21
> **Response to Reviewer kpcU**
>
> Thanks for your insightful suggestions. Below, please find the responses to some specific comments.
>
> ---
>
> **Point 1:** *Discussion on over-smoothing and over-squashing issues is needed.*
>
> **Reply 1:** We agree with you and have added a discussion in Section A.6 of the revised manuscript. As far as we know, whether HGNNs suffer from over-smoothing and over-squashing issues is still an open question. We think it can be discussed in two situations.
>
> HGNNs can be divided into two categories, i.e., metapath-free HGNNs and metapath-based HGNNs. The former directly incorporates the heterogeneity information in representation learning using a GNN-like version. So, these methods, e.g., HINormer, need to solve the over-smoothing and over-squashing issues. The latter selectively aggregates messages with the assistance of manually designed or automatically generated meta-paths. The number of employed meta-paths typically does not increase with the hops, and each target node only incorporates a part of effective information. Consequently, these methods are typically not affected by over-smoothing and over-squashing issues.
>
> For example, in our method, each target node only aggregates a part of neighborhood information with the guidance of effective meta-paths. It means the farther a node is from the target node, the less likely it is to be aggregated. So, each target node incorporates different information and the over-smoothing issue is at least alleviated. Additionally, we set the number of selected meta-paths $M = 30$ for all datasets. $M$ is a constant independent of the maximum length/hops. So, the over-squashing issue does not exist.
>
> ---
>
> **Point 2:** *Some discussion on the advantages and disadvantages of transformers, especially HINormer, compared to this work should be elaborated*.
>
> **Reply 2:** In Table 11 of the Appendix, we have shown the comparison with transformers. Specifically, we replace the concatenation operation in LMSPS with the transformer module for semantic attention. We can see that the transformer block can not improve performance and spends more training time, parameters, and GPU memory. HINormer is currently the best transformer-based HGNN. Based on Table 3, our LMSPS shows better performance than HINormer on small HIN datasets. In addition, HINormer does not show the results on large datasets like representative ogbn-mag, while our LMSPS ranks top-1 on the leaderboard of Open Graph Benchmark from Sep 29, 2023 to Oct 24, 2023. Because HINormer has not provided the source code until now, we can not compare the training efficiency with it. However, SeHGNN is also a transformer-based HGNN. We have compared the time complexity, effectiveness, efficiency, and scalability of SeHGNN and LMSPS in Sections 5.2 and 6. LMSPS has an obvious advantage.
>
> We agree with you that transformers have the potential to capture long-distance dependencies in HINs. The main advantage of transformers compared to our MLP-based model is their greater potential to become cross-field unified models. Following your suggestion, we have added more discussion on transformers in Section A.8 of the revised manuscript. Thanks.
>
> ---
>
> **Point 3:** *Writing is not very clear, especially in Section 5.1. It is more like a step-by-step introduction to the pipeline.*
>
> **Reply 3:** Sorry for the confusion. Although Reviewer vDix and f416 think our paper is well written and the presentation is clear, we admit Section 5.1 can be improved. We have rewritten Section 5.1 in the revised manuscript to make our contributions and focuses more clear.
>
> Because our search space contains all target-node-related meta-paths, severely challenging the effectiveness and efficiency of searching, Section 5.1 introduces two novel strategies for efficiently searching effective meta-paths.
>
> To overcome the efficiency challenge,  we use a $progressive$ $sampling$ search algorithm. The $progressive$ means the search space size is progressively shrunk from $K$ to $2M$ based on the learned path strength. The $sampling$ means we sample $M$ meta-paths from dynamic search space in each iteration for parameter update to improve efficiency.
>
> To overcome the effectiveness challenge, we sample $M$ meta-paths from the retained $2M$ meta-paths for forward propagation and evaluate the validation loss. The $sampling$ $evaluation$ is repeated multiple times. Our final search meta-paths are the $M$ meta-paths with the lowest validation loss.
>
> ---
>
> **Point 4:** *Minor issue: In Figure 2, search stage, I'm not sure if the example 0/1 values given (in the green block) are right.*
>
> **Reply 4:** Sorry for the confusion. It is right. Because the whole search algorithm is very complex, we use the green block to illustrate the sampling process in the super-net training. The sampling process uses a uniform sample, so the 0/1 values are independent of the path strength. We have clarified it in Figure 2 of the revised manuscript. Thank you!

---

> ### Comment · Reviewer_kpcU · 2023-12-03
>
> While some responses are detailed and informative, some are hand-waving (points 1 and 2).
> My score remains unchanged, but I would not mind if the paper is accepted.

---

### Official Review · Reviewer_zWVu · 2023-11-06

**Soundness:** 4 excellent
**Presentation:** 3 good
**Contribution:** 3 good
**Rating:** 8
**Confidence:** 3

**Summary:**

This paper aims to solve meta-path instance selection for heterogeneous graphs. Two observations of existing works trigger the model design, including (1) only several meta-path instances dominate the model performance, and (2) some meta-path instances cause a negative impact on the performance. Therefore, the paper proposes a search algorithm to search meta-path instances from all possible instances and use simple MLP to predict labels after finishing searching paths. Compared with meta-path-free models, meta-path-based models, and NAS-based models, the proposed model consistently performs the best.

**Strengths:**

1. Motivation is very clear and the experimental results on effectiveness and efficiency support major claims.
2. the architecture of the model is very simple.
3. the empirical results show a strong performance.

**Weaknesses:**

1. The maximum path length is pre-defined still.
2. Comparison with existing works on the number of parameters should be also added to show how powerful MLP is.
3. The contribution of each meta-path instance is not clear. Will the searched long-range path have a negative impact on performance?

**Questions:**

1. The embeddings of the target nodes are all learned from neighbors, why not use the node features of the target nodes?
2. How do you set up and learn $\alpha_k$ for all meta-path instances?
3. Why do you think 2M is a good number for the candidates to be searched?

---

> ### Author Response · Authors · 2023-11-21
> **Response to Reviewer zWVu**
>
> Thanks for your positive comments that greatly encourage us. In the following, we respond to your concerns point by point.
>
> ---
>
> **Point 1:** *Comparison with existing works on the number of parameters should be also added to show how powerful MLP is.*
>
> **Reply 1:** Figure 4 in the Appendix of the manuscript has shown the comparison with existing works on the number of parameters. Thanks.
>
> ---
>
> **Point 2:** *The contribution of each meta-path instance is not clear. Will the searched long-range path have a negative impact on performance?*
>
> **Reply 2:** As we stated in Section A.5, i.e., interpretability of searched meta-paths, because we discover many more meta-paths than traditional methods and most meta-paths are longer than traditional meta-paths, it is tough to interpret and explore their contribution one by one. In Section A.5, we focus on the interpretability of meta-paths on obgn-mag from the Open Graph Benchmark.
>
> Yes, as we discussed in Section 6.7, the performance of LMSPS does not always increase with the value of the maximum hop, and the best maximum hop depends on the sparsity of the dataset. In Table 4, we conduct experiments to demonstrate that longer meta-paths are more useful for sparser HINs.
>
> ---
>
> **Point 3:** *Why not use the node features of the target nodes?*
>
> **Reply 3:** Actually, the embeddings of the target nodes are learned from both neighbors and target nodes. As shown in Figure 2, our candidate meta-paths contain the 0-hop meta-path A, i.e.,  the type node itself. Based on Eq. (1), the neighbor aggregation process for $l$-hop meta-path $P_k=cc_1c_2\ldots c_l$ is $X_k=A_{c,c_1}A_{c_1,c_2}\cdots A_{c_{l-1},c_l}X^{c_l}$, where $A_{c_i,c_i+1}$ be the row-normalized adjacency matrix, $c_l$ is the target node type and $X^{c_l}$ is the node features of the target nodes. When $l=0$, $  X_k=X^{c_l}$, i.e., the raw feature of the target nodes. So, the embeddings of the target nodes are learned from both neighbors and target nodes. We have clarified it after Eq. (1) in the revised manuscript.
>
> ---
>
> **Point 4:** *How do you set up and learn $\alpha_k$ for all meta-path instances?*
>
> **Reply 4:** Sorry for the confusion. As shown in Section 5.1, $\alpha =\{\alpha_1,\cdots,\alpha_k, \cdots,\alpha_K\} \in \mathbb{R}^K$  corresponds to the architecture parameters of candidate meta-paths $\mathbb{P}=\{P_1,\cdots,P_k,\cdots,P_K\}$.  So for each candidate meta-path $P_k$, there is a corresponding $\alpha_k$ to learn its path strength. All $\alpha_k$s are initialized as $1$s to keep the same initial path strength. In the search stage, we freeze architecture parameters $\alpha$ when training $\omega$ on the training set and freeze $\omega$ when training $\alpha$ on the validation set. These two update processes are performed in an alternative manner.  We have clarified it more clearly in Sections 5.1 and 6.2 of the revised manuscript.
>
> ---
>
> **Point 5:** *Why do you think $2M$ is a good number for the candidates to be searched?*
>
> **Reply 5:** We set the number of searched meta-paths to be $M$. To search $M$ effective meta-paths, there are two steps. First,  we use a progressive sampling search algorithm to shrink the search space size from $K$ to $2M$. Then, we sample $M$ meta-paths from $2M$ meta-paths for forward propagation and evaluate the validation loss. The sampling evaluation is repeated $200$ times. Our final searched meta-paths are the $M$ meta-paths with the lowest validation loss. So $V=2M$ is a parameter to trade off the importance of progressive sampling search and sampling evaluation. When $V$ is too large, we need to repeat the sampling evaluation many more times, which will decrease the efficiency. When $V$ is too small, some effective meta-paths may be dropped too early. So, we set $V=2M$.

---

### Meta-Review · Area_Chair_b76q · 2023-12-05

**Metareview:**

The paper studies the problem of identifying meta-paths in heterogeneous graphs. In particular a new method, called Long-range Meta-path Search through Progressive Sampling, is introduced to detect meta-paths by narrowing the search space by leveraging characteristics of the meta-paths.

The paper contains some interesting ideas and some interesting experimental insights although during the discussion phase few important concerns have been raised by the reviewers:

- the writing of the paper is too dense and the paper is a bit hard to read

- the paper is missing some important discussion(for example about oversmoothing and over-squashing or alternative transformer methods)

- the motivations for looking for long range meta-paths are not too convincing

Overall, the paper has certainly some merit but it is not clear that it is above the ICLR acceptance bar.

**Justification For Why Not Higher Score:**

- the writing of the paper is too dense and the paper is a bit hard to read

- the paper is missing some important discussion(for example about oversmoothing and over-squashing or alternative transformer methods)

- the motivations for looking for long range meta-paths are not too convincing

**Justification For Why Not Lower Score:**

N / A

---

### Decision · Program_Chairs · 2024-01-16

Reject